# The deletion of a major facilitator superfamily gene *VdMFS2* results in enhanced pathogenicity of *Verticillium dahliae* to cotton

Tiange Sun,[1] Yongtai Li,[1] Yuanjing Li,[1] Ruixiang Yuan,[1] Feng Liu,[1] Xiaomei Ma,[2] Xinyu Zhang,[1] Yuqiang Sun,[3] Yanjun Li,[1] Jie Sun[1]

**ABSTRACT**    Among soil-borne phytopathogenic fungi, *Verticillium dahliae* is recognized as a devastating causal agent of cotton Verticillium wilt. Elucidation of molecular interactions between *V. dahliae* and cotton plants remains critical for disease control strategy development. A highly expressed major facilitator superfamily (MFS) transporter gene (VDAG_09088, designated *VdMFS2*) was identified through previous RNA sequencing (RNA-seq) analysis of *V. dahliae* exposed to cotton root exudates. Functional characterization was performed using host-induced gene silencing (HIGS), targeted gene knockout, and heterologous expression in *Saccharomyces cerevisiae*. Additionally, the secreted proteins of wild-type and *VdMFS2* deletion mutant (Δ*VdMFS2-4*) were compared. It was found that *VdMFS2* silencing enhances the pathogenicity of *V. dahliae* to cotton. The Δ*VdMFS2-4* and Δ*VdMFS2-5* mutants displayed accelerated colony growth rates, enhanced carbon source utilization ability, reduced leucine (Leu) and valine (Val) utilization ability, increased stress tolerance, increased host penetration ability, and accompanied by elevated virulence. Proteomic profiling identified 94 upregulated secreted proteins in the mutant, including 54 canonically secreted proteins (CSPs) and 40 non-canonically secreted proteins (N-SCPs). Notably, 23 carbohydrate-active enzymes (CAZymes) and 18 small cysteine-rich proteins (SCRPs) were detected among the CSPs, demonstrating collective contributions to pathogenicity enhancement. Taken together, *VdMFS2* was demonstrated to function as a negative growth, development, and stress resistance regulator of *V. dahliae*, capable of transporting certain amino acids. Gene disruption triggered compensatory secretion of virulence effectors, particularly CAZymes and SCRPs, ultimately leading to enhanced virulence of *V. dahliae*.

**IMPORTANCE**    The knockout of *VdMFS2* can inhibit the proliferation of mycelia and the germination of spores of *V. dahliae* while enhancing its pathogenicity to cotton. This study provides new insights into the role of major facilitator superfamily (MFS) transporters in *V. dahliae* and offers more precise clues for elucidating the impact of MFS transporters on this fungus.

**KEYWORDS**    *Verticillium dahliae*, cotton, *VdMFS2* transporter, host-induced gene silencing (HIGS), gene knockout, secreted proteins

Cotton is one of the most important cash crops in the world. It is a major source of natural fiber, and its seed is also an important material for cooking oil. In the growing process of cotton, it is often harmed by various diseases and insects. Verticillium wilt (VW) is known as the "cancer" of cotton, which seriously affects the yield and quality of cotton and causes great economic losses all over the world (1, 2). VW is caused mainly by *Verticillium dahliae*, which is a soil-borne plant pathogenic fungus that can cause wilting and necrosis of many important crops (3). When under stress or without a

**Peer Reviewer** Xiaofeng Su, Chinese Academy of Agricultural Sciences, Beijing, China

Address correspondence to Yuqiang Sun, sunyuqiang@zstu.edu.cn, or Yanjun Li, liyanjun@shzu.edu.cn.

Tiange Sun and Yongtai Li contributed equally to this article. The order of the authors was determined based on their overall contributions to the study design, experimental implementation, data analysis, and manuscript writing, among which Tiange Sun led the core experimental work and the drafting of the manuscript's first draft.

The authors declare no conflict of interest.

suitable host, this fungus produces melanized microsclerotia that can survive in the soil for up to 20 years until the conditions are suitable for infection (1, 4). In recent years, VW has become increasingly serious due to climatic variation, long-term monoculture, and frequent introduction of new cotton varieties/hybrids in various countries and regions in the world (5). At present, due to the complex pathogenic mechanism of *V. dahliae*, there is still a lack of effective measures for the prevention and control of VW.

Pathogenic fungi need to use various metabolites secreted by host to provide nutrition and energy in the infection process. Transport systems play a critical role in the export of secondary metabolites and waste compounds during infection (6). In fungi, ATP-binding cassette (ABC) and the major facilitator superfamily (MFS) transporter proteins are two main classes of transporters (7). MFS family is one of the largest membrane transport protein families widely existing in living organisms. Its basic function is to assist in the transmembrane transport of various substances. Most member proteins of the MFS superfamily consist of 400 to 600 amino acid residues, with both N and C termini located in the cell. The prediction of the secondary structure indicated that most of the proteins had 12 α-helical transmembrane domains (8), and others with 14 or 24 alpha-helices that may have evolved on the basis of 12 transmembrane α-helices. This unique way of folding is also named "MFS fold" (9–11), which enables its transporters to transport a wide range of substances.

MFS family proteins can transport monosaccharides, polysaccharides, amino acids, peptides, vitamins, enzyme cofactors, drug molecules, chromophores, bases, and many other small molecules (12–15). In some cases, MFS is responsible for delivering nutrients, particularly sugars, to cells, which provides an advantage for fungal growth (16). The knockout or silencing of MFS family genes in some fungi has an effect on the fungal growth, development, and virulence (3, 17, 18). For example, knockout of FgMFS1 significantly reduced the pathogenicity of the Fusarium graminearum on wheat (19). In *V. dahliae*, knockout of the MFS family gene *VdPAT1* attenuates its pathogenicity to cotton (20). Additionally, some MFS family genes have been reported to be involved in stress response. For example, AaMFS19 in *Alternaria alternata* was crucial for the fungal resistance to oxidative stress and fungicides. Mutants lacking AaMFS19 were more sensitive to certain chemicals (21). In *Candida albicans*, MFS transporters were involved in resistance to a variety of antifungal drugs. These transporters contribute to drug resistance by extruding the drugs, helping the fungal cells to eliminate harmful drug molecules (22).

The MFS transporter genes in *V. dahliae* have not been reported yet. In our previous study, RNA-seq was used to analyze the gene expression of *V. dahliae* after sensing root exudates from different cotton varieties (23). It was found that the expression level of an MFS transporter gene (VDAG_09088), named as *VdMFS2*, increased 10.9-fold after induction by root exudates from a susceptible cotton variety, suggesting that it may be required for the pathogenicity of *V. dahliae*. In this study, the *VdMFS2* gene was knocked out to obtain its deletion mutants and investigate the growth rate, hyphal morphology, sporulation, and pathogenicity of the mutants. We then conducted comparative proteomic analysis of the secreted proteins of the Δ*VdMFS2* mutant before and after induction by cotton root tissues to explore the reasons for the altered pathogenicity of the mutant. This research deepens our understanding of the role of *VdMFS2* during the interaction between cotton and *V. dahliae* and provides key information for developing effective strategies against VW in cotton.

## MATERIALS AND METHODS

### Fungal strain, plant material, and culture conditions

The wild-type *V. dahliae* strain Vd592 is a highly pathogenic and defoliating strain. The fungus was maintained on potato dextrose agar (PDA) medium at 25°C in the dark. Upland cotton cultivar "Xinluzao 7" is susceptible to *V. dahliae* and was used in this

study. Cotton seeds were grown in a plant incubator at 28°C with a photoperiod of 16-h light/8-h dark and a relative humidity of 60%.

## Bioinformatics analysis of *VdMFS2*

The GFF3, CDS (Coding sequence), genomic and protein sequence files of *V. dahliae* ASM15067v2 were downloaded from NCBI (https://www.ncbi.nlm.nih.gov/). Phylogenetic analysis of *VdMFS2* and other major facilitator superfamily (MFS) transporters was conducted employing the MEGA-X software with the neighbor-joining method. Multiple sequence alignment of the MFS transporter genes in *V. dahliae* was performed using MEGA-X software. Prediction of the transmembrane domains of *VdMFS2* was facilitated by the online TMHMM-2.0 server (https://services.healthtech.dtu.dk/service.php?TMHMM-2.0). Additionally, signal peptides of *VdMFS2* were predicted using the same online resource.

## Host-Induced gene silencing (HIGS) treatment of *VdMFS2*

The total RNA of *V. dahliae* Vd592 was extracted using the Fungal RNA Kit (Omega Inc., Guangzhou, China) and transcribed into cDNA using TransScript One-Step gDNA Removal and cDNA Synthesis SuperMix. The *VdMFS2* interference fragment (314 bp) was amplified from Vd592 cDNA using the HIGS-*VdMFS2*F/R primers (Table S1) and then cloned into the pTRV2 vector. The pTRV2-*VdMFS2*, pTRV1, empty vector pTRV2-*00,* and positive control vector pTRV2-*GhCHLI* (the gene mutation causes the leaves to lose their color) plasmids were transformed into the *Agrobacterium tumefaciens* strain GV3101 using the freeze-thaw method (24), respectively. The *Agrobacterium* suspensions were prepared with MES (2-(N-Morpholino) ethanesulfonic acid) according to previous report (25). The suspensions containing pTRV2-*GhCHLI*, pTRV2-*VdMFS2*, and pTRV2-*00* were each mixed with the suspension containing pTRV1 at a 1:1 ratio and incubated at room temperature for 3 h prior to injection into cotton seedlings. The suspensions were injected into the flattened cotyledons of 10-day-old cotton seedlings by the compression method with a 1-mL needleless syringe until the cotyledons were filled with the suspension. Each suspension containing pTRV2-*00* and pTRV2-*VdMFS2* was injected into 50 cotton seedlings, respectively. The injected cotton seedlings were then placed in the dark at 22°C–25°C for 24 h and subsequently cultured under normal light conditions (26).

## Assessment of disease resistance in cotton

When the TRV-treated cotton seedlings reached the two-true-leaf stage, the spore suspension of *V. dahliae* Vd592 was diluted to $1 \times 10^7$ CFU/mL and inoculated into the roots of cotton plants through the root-injury method (27). Disease symptoms, including vascular bundle browning and the cotton disease index, were assessed at 14 and 28 dpi (days post-inoculation). The severity of disease symptoms in infected plants was graded on a scale from 0 to 4, as previously reported (28). The disease index (DI) was calculated using the formula: DI = [(Σ disease grade × number of infected plants) / (total number of sampled plants × 4)] × 100. At 14 dpi, the infected stems were longitudinally cut using a sterile scalpel, and the vascular discoloration was observed and photographed. For the fungal recovery assay, infected stems were collected at 28 dpi, sterilized, and then cut into segments of 1 cm. The stem segments were cultivated on PDA medium and incubated at 25°C in the dark. The fungal mycelia generated from the stem segments were observed and photographed after 5 days of incubation. To ascertain fungal biomass and the efficiency of gene silencing, roots and stems from cotton plants from pTRV2-*00* and pTRV2-*VdMFS2* treatments were harvested at 21 dpi for DNA and RNA extraction. Total DNA was extracted using the CTAB method, and RNA was isolated using the EASYspin Plus Plant RNA Extraction Kit (Aidlab, Beijing, China). For fungal biomass detection, qRT-PCR was conducted with cotton root and stem DNA as templates, *Ve-ITS1*-F/ *ST-VE1*-R as primers, and *GhUBQ7* (DQ116441.1) as endogenous reference genes (Table S1). The expression of *VdMFS2* was quantified by

qRT-PCR, utilizing cotton root and stem cDNA as templates, *VdMFS2*-qF/qR as primers, and *V. dahliae β-tubulin* (*VDAG_10074*) as internal reference gene (Table S1).

## Gene knockout and complementation

In order to construct the *VdMFS2* gene deletion vector, a gene disruption structure was designed to target the *VdMFS2* gene. The hygromycin resistance gene (Hyg) (1,878 bp) was used to replace the target fragment (937 bp) of the gene to be deleted. The upstream (976 bp) and downstream (1,060 bp) fragments of the *VdMFS2* gene were amplified by PCR using the primer pairs *VdMFS2*-Flank-5F/R and *VdMFS2*-Flank-3F/R, respectively (Table S1). The hygromycin resistance (Hyg) fragment was amplified from the T-HPH plasmid using the primer pair *VdMFS2*-Hyg-F/R (Table S1). Subsequently, the three amplified fragments were recombined into the pGKO$_2$-Gate knockout vector using the ClonExpress II One Step Cloning Kit (Vazyme Biotech Co., Ltd., Nanjing, China). The recombinant vector was transformed into Vd592 through *Agrobacterium*-mediated transformation. Transformants were screened on PDA medium containing 50 mg/mL hygromycin and confirmed by PCR and qRT-PCR. The qRT-PCR primers were *VdMFS2*-qPCR-F/R (Table S1). The qRT-PCR was performed using SYBR Premix Ex Taq (TakaRa) on a LightCycler 480 System II (Roche, United States) instrument. The relative expression ratio of each gene was calculated from the cycle threshold (Ct) values (29). The *β-tubulin* gene (β-tubulin F/R) of *V. dahliae* (*VDAG_10074*) was used as an endogenous control.

For the construction of the complementation vector, a 3,841-bp fragment containing the promoter, coding region, and terminator sequences of *VdMFS2* was amplified from the Vd592 genomic DNA using the primers *VdMFS2*-C-F/R (Table S1). The amplified fragment was then cloned into the pSULPH-mut-PG#PB vector using the ClonExpress II One Step Cloning Kit. The recombinant vector was transformed into the *VdMFS2* deletion mutant through *Agrobacterium*-mediated transformation. Transformants were confirmed by PCR and qRT-PCR.

## Fungal growth assays

The wild type (Vd592), deletion mutants (Δ*VdMFS2-4* and Δ*VdMFS2-5*), and complementary mutants were inoculated into liquid CM medium (Complete medium) and incubated at 25°C for a week. Conidia from different strains were collected and were diluted to $1 \times 10^7$ CFU/mL with water. Then 10 µL of conidial suspension was dropped onto PDA, CM, and Czapek Dox medium and incubated at 25°C. PDA medium is a basic medium, which includes comprehensive and balanced nutrition that effectively meets the nutritional needs for microbial growth and development. CM culture medium contains various vitamins and trace elements that are beneficial for fungal growth. Czapek-Dox (Defined Medium): Contains only 0.02% KNO$_3$ as the nitrogen source. For plant-pathogenic fungi, such as *V. dahliae*, low-nitrogen stress is a critical trigger for microsclerotia formation; Czapek-Dox medium thus allows simultaneous comparison of differentiation rate and microsclerotial yield between the wild-type and mutant strains (29). The colony diameter was measured and photographed at 15 days post-inoculation. The conidial concentration of each strain was adjusted to $1 \times 10^7$ CFU/mL using a hemacytometer and was then inoculated into liquid CM medium cultured on a shaker (180 rpm/min) at 25°C. The conidial yield was determined using a blood cell counting plate under a microscope every day. Conidia from different strains were diluted to $1 \times 10^3$ CFU/mL and were then evenly distributed on PDA medium using a triangular spreading rod. After 24 h of incubation, the monoconidial growth was observed under a microscope. Each strain was replicated at least three times.

## Fungal pathogenicity assays

For the mycelial penetration assay, 10 µL of conidial suspension ($1 \times 10^7$ CFU/mL) from each strain was dropped onto a sterilized carboxymethyl cellulose membrane overlaid on PDA medium and incubated in the dark at 25°C for 7 days. After removing the

carboxymethyl cellulose membrane, the PDA medium was incubated for another 3 days, and the colonies of each strain were observed and photographed. The experiment was conducted three times.

When the two true leaves of "Xinluzao 7" unfolded, the spore suspensions of the wild-type (Vd592), deletion mutants (ΔVdMFS2-4 and ΔVdMFS2-5), and complemented mutants were diluted to $1 \times 10^7$ CFU/mL and inoculated into cotton plants via the root-injury method (24). At 14 and 28 days post-inoculation (dpi), disease symptoms, including vascular bundle browning and the cotton disease index, were assessed. The disease resistance assessment of the plants was performed as described above.

## Carbon source and nitrogen source utilization assays

To test the carbon source utilization capacity of each strain, 10 µL of conidial suspension ($1 \times 10^7$ CFU/mL) was dropped onto the center of Czapek-Dox medium containing 1% glucose, sucrose, xylose, pectin, cellulose, respectively. Czapek-Dox medium without carbon sources was used as the control. The colony diameter of each strain was measured and photographed at 15 days post-inoculation. The experiment was repeated three times.

To test the nitrogen source utilization capacity of each strain, 10 µL of conidial suspension ($1 \times 10^7$ CFU/mL) was dropped onto the center of Czapek-Dox medium containing 1% Phe (Phenylalanine), Leu (Leucine), Val (Valine), Cys (Cysteine), and Trp (Tryptophan), respectively. Czapek-Dox medium without nitrogen sources was used as the control. The colony diameter of each strain was measured and photographed at 15 days post-inoculation. The experiment was repeated three times.

## Heterologous expression of *VdMFS2* in yeast

The open reading frame (ORF) of *VdMFS2* was amplified using primers PDR-VdMFS2-F/R and cloned into the pDR195 vector at the *Xho* I and *BamH* I restriction sites (Table 1). The resulting construct was then transformed into the hexose transporter-deficient yeast strain EBY.VW4000. This strain, constructed by Wieczorke (30), lacks all 18 hexose transporter genes (*HXT1–HXT17* and *GAL2*) as well as the two glucose sensors *SNF3* and *RGT2*. Consequently, it cannot utilize hexoses (e.g., glucose, fructose, mannose) as a sole carbon source and is widely used as a reporter strain for functional studies of hexose transporters. To assess function, cell suspensions ($1 \times 10^7$ CFU/mL) were serially diluted (10-fold each) to concentrations of $10^7$, $10^6$, $10^5$, and $10^4$ CFU/mL. Aliquots (6 µL) from each dilution were spotted onto solid SD medium containing glucose, maltose, galactose, sorbitol, sucrose, or xylose as the sole carbon source. All experiments were performed in triplicate.

## Stress response assays

To test the tolerance of each strain to stress, 10 µL of conidial suspension ($1 \times 10^7$ CFU/mL) of each strain was dropped onto the center of PDA medium containing 5 mM $CuSO_4$, 0.02% Congo red (CR), 1 M NaCl, 0.002% SDS, and 30 µg/mL calcofluor white (CFW), respectively. PDA medium without stress inhibitors was used as the control. The colony diameter of each strain was measured and photographed after 15 days of inoculation. The inhibition rate of the colonies was calculated using the formula: inhibition rate (%) = [(control growth diameter − treatment growth diameter)/control growth diameter] × 100.

To evaluate carbendazim tolerance, a conidial suspension (100 µL; $1 \times 10^7$ CFU/mL) of each strain was evenly spread on PDA plates. Sterile filter-paper discs (8 mm in diameter) were soaked in carbendazim solutions at concentrations ranging from 0.5 to 10 g/L and then aseptically placed onto the inoculated agar surface. PDA plates without carbendazim served as controls. After 15 days of incubation, the diameter of the inhibition zones was measured and photographed, and the inhibition rate was calculated. The experiment was performed with three replicates.

## Secreted protein content determination

The spores of different strains were adjusted to $1 \times 10^7$ CFU/mL and cultured in 200 mL of liquid Czapek medium for 12 days. The culture liquid was filtered through a 0.22-μm filter to obtain the total protein solution of each strain. The $(NH4)_2SO_4$ solution was added to protein solution. After centrifugation, the supernatant was discarded, and the protein precipitate was dissolved in 500 μL of PBS buffer solution. The total protein was quantitatively determined using the Coomassie brilliant blue protein assay kit (AIDISHENG Inc., Shanghai, China). The experiment was repeated three times.

## Reactive oxygen species (ROS) burst analysis

Cotton seedlings of "Xinluzao 7" were cultivated in a light incubator with an appropriate amount of water. Several small holes were punctured on both sides of the true leaves using a syringe. A total secreted protein sample of 1.5 μL was applied to the small holes on the leaves. After culturing for 48 h, the leaves were removed and immersed in diaminobenzidine (DAB) dye solution (Coollaber INC., Beijing, China) for 8 h in the dark. The leaves were decolorized with absolute ethanol to observe the brown pigment deposition (DAB). Determination of $H_2O_2$ concentration in cotton leaves treated with protein solutions from different strains using the xylenol orange method (31).

## TMT labeling, spectrometric analysis, and data processing

The wild-type (WT) and ΔVdMFS2-4 strains were co-cultured with root tissues of "Xinluzao 7" in Czapek-Dox medium for 5 days, respectively, while the two strains were also cultured separately in Czapek-Dox medium. The samples cultured in Czapek-Dox medium were designated as Vd592 and ΔVdMFS2-4, and those co-cultured with root tissues were designated as Vd592Y and ΔVdMFS2-4Y. Secreted proteins were collected and precipitated for proteomic sequencing. Protein extraction and digestion, TMT labeling, peptide identification, and protein identification were all performed by Biomarker Technologies Co., Ltd. (Beijing, China), as described by Ross (32). Differentially expressed genes were screened using a fold change greater than 1.2 and an FDR less than 0.01 as the threshold (32). GO functional enrichment analysis was performed using the R/top GO (2.18.0) method, with a $P$-value less than 0.05 as the threshold.

## Enzyme activity assay

The spore suspensions of four strains were adjusted to $1 \times 10^7$ CFU/mL, and 100 μL of each spore suspension was added to 50 mL Czapek-Dox liquid medium containing 0.1% methyl cellulose, xylan, and pectin, respectively. The mixtures were cultured in the dark at 200 r/min for 7 days. The absorbance at 540 nm of different strains in different media was measured using the DNS method (33). The activities of cellulase, xylanase, and pectinase were calculated using the formula (U/mL) = (C × V × $10^6$)/(W × t × f). The experiment was repeated three times.

## RESULTS

### Bioinformatics analysis

*VdMFS2* (VDAG_09088) is 1,841 bp in length with six introns, and its open reading frame (ORF) is 1,470 bp. The deduced peptide has 489 residues and 10 transmembrane domains (TMDs) (Fig. 1A). MFS transporters can be classified into 17 subfamilies according to previous reports (19). It can be known through evolutionary relationships that VdMFS2 belongs to the ACS (anion: cation symporter) subfamily in MFS superfamily (Fig. 1B). Members of the ACS family harbor seven residues (GEPWPER) of the characteristic motif in the fourth transmembrane domain (34). The multiple sequence alignment discovered that *VdMFS2* contained four conserved amino acids (GEPE) the same as the reported residues, speculating that there may have been some variations in the MFS genes in *V. dahliae* during species evolution.

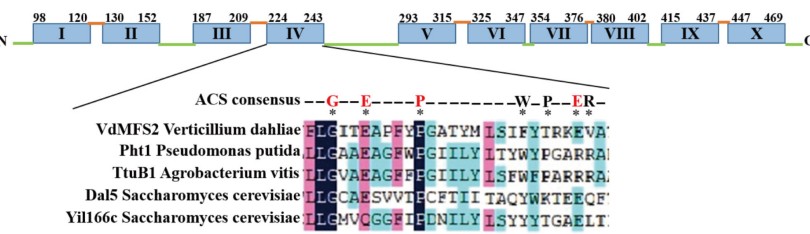

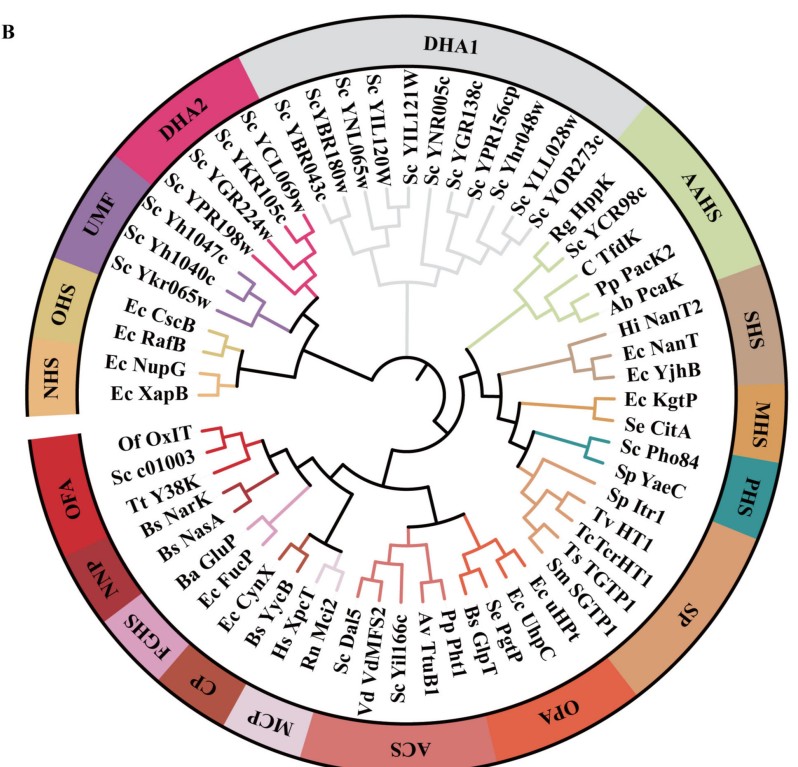

**FIG 1** Bioinformatics analysis. (A) Predicted transmembrane domains in *VdMFS2* proteins. Turquoise boxes indicate transmembrane domains, green lines indicate cytosolic regions, and orange lines indicate extracellular regions. Multiple sequence alignment of the 4th transmembrane domain in five proteins of the ACS subfamily, with amino acid identity to the ACS consensus sequence indicated by fuchsia. (B) Phylogenetic analysis of MFS superfamily. MFS transporters can be classified into 17 families: DHA1 (drug: H+ antiporter [12-TMS] drug efflux family), ACS (anion: cation symporter) family, NNP (nitrate–nitrite porter) family (includes NasA and NarK from *Bacillus subtilis*), OPA (organophosphate: inorganic phosphate antiporter) family, AAHS (aromatic acid: H+ symporter) family, SP (sugar porter) family, PHS (phosphate: H+ symporter) family, MHS (metabolite: H+ symporter) family, SHS (sialate: H+ symporter) family, OFA (oxalate: formate antiporter) family, MCP (monocarboxylate porter) family, CP (cyanate permease) family, FGHS (fucose–galactose–glucose: H+ symporter) family, OHS oligosaccharide: H+ symporter) family, NHS nucleoside: H+ symporter) family, UMF (unknown major facilitator) family, and DHA2 (drug: H+ antiporter [14-TMS]) drug efflux family. Represented species: Sc (*Saccharomyces cerevisiae*), Rg (*Rhodococcus globerulus*), C (*Cupriavidus*), Pp (*Pseudomonas putida*), Ab (*Acinetobacter baylyi*), Hi (*Haemophilus influenzae*), Ec (*Escherichia coli*), Sp (*Schizosaccharomyces pombe*), Tv (*Trypanosoma vivax*), Tc (*Trypanosoma cruzi*), Ts (*Taenia solium*), Sm (*Schistosoma mansoni*), Se (*Salmonella enterica*), Bs (*Bacillus subtilis*), Av (*Agrobacterium vitis*), Rn (*Rattus norvegicus*), Hs (*Homo sapiens*), Ba (*Brucella abortus*), Tt (*Thermoproteus oteustenax*), Of (*Oxalobacter formigenes*).

## HIGS of *VdMFS2* led to decreased cotton resistance to VW

Host-induced gene silencing (HIGS) technology was used to silence *VdMFS2* to explore the impact of the gene silencing on cotton resistance to VW. When susceptible upland cotton variety (Xinluzao 7) reached the two-true leaf stage, the Vd592 strain was inoculated onto both pTRV2-*00* and pTRV2-*VdMFS2* (pTRV2-*VdMFS2-1* and pTRV2-*VdMFS2-2*) cotton plants. The cotton disease symptoms were examined at 14 and 28 days post inoculation (dpi). Compared with pTRV2-*00*, the pTRV2-*VdMFS2* plants exhibited more severe disease symptoms, with significant yellowing and even shedding of leaves. The vascular bundles of pTRV2-*VdMFS2* plants showed more severe browning, with a larger amount of isolated *V. dahliae* and a higher fungal biomass (Fig. 2A and B). The disease index of pTRV2-*VdMFS2* plants was significantly higher at both time points (14 and 28 dpi) than that of pTRV2-*00* plants (Fig. 2C). The expression level of *VdMFS2* in pTRV2-*VdMFS2* plants was significantly reduced compared with pTRV2-*00* plants (Fig. 2D), indicating that the expression of *VdMFS2* was successfully suppressed. The chlorophyll content and the activity of superoxide dismutase (SOD) enzyme in leaves of pTRV2-*VdMFS2* plants were significantly decreased (Fig. 2E and F), while malondialdehyde (MDA) content was elevated (Fig. 2G), suggesting that the pTRV2-*VdMFS2* plants suffer greater damage compared with those pTRV2-*00* plants. Taken together, these results showed that *VdMFS2* silencing by HIGS leads to an increase in fungal biomass accumulation and an enhanced pathogenicity of *V. dahliae* to cotton.

## *VdMFS2* deletion accelerated the growth of *V. dahliae*

The *VdMFS2* was knocked out in the wild-type (WT) Vd592 genome using homologous recombination mediated by a PEG-mediated transformation method. Two independent deletion mutants (Δ*VdMFS2-4* and Δ*VdMFS2-5*) were obtained and verified by PCR and qRT-PCR. In addition, complementary mutants (Δ*VdMFS2-C1* and Δ*VdMFS2-C2*) were obtained in the background of knockout mutants by reintroduction with the *VdMFS2* copy and confirmed by PCR and qRT-PCR.

To investigate the differences among WT, deletion mutants, and complementary strains, spore suspensions were prepared and inoculated on CM, Czapek-Dox, and PDA media. After 15 days of culture, the colony size of the Δ*VdMFS2* mutants was found to be significantly larger than those of the WT and Δ*VdMFS2-C*. The germination process of single spores cultured on PDA medium was observed. It was found that the hyphae of the Δ*VdMFS2* mutants grew faster than those of WT and Δ*VdMFS2-C*. At 24 h after inoculation, the hyphal length of Δ*VdMFS2* mutants was significantly longer than that of WT and Δ*VdMFS2-C* (Fig. 3A and B). The spore yield of the Δ*VdMFS2* mutants was significantly higher than that of WT and Δ*VdMFS2-C* after 6 days of cultivation (Fig. 3D). These results suggested that the *VdMFS2* deletion led to accelerated growth of *V. dahliae*.

## *VdMFS2* deletion led to increased pathogenicity of *V. dahliae*

To determine the role of *VdMFS2* in the pathogenicity of *V. dahliae*, the hyphal penetration ability of Vd592, Δ*VdMFS2* mutants, and complementary strains through the cellophane membrane was compared. Different strains were cultured for 7 days on cellophane membrane covering PDA. At 3 days after removing the cellophane membrane, two *VdMFS2* deletion mutants exhibited obviously larger colonies grown from the hyphae penetrating through the membrane compared to Vd592 and complementary strains. The results indicated that *VdMFS2* deletion enhanced the ability of *V. dahliae* to penetrate plant cell walls (Fig. 4A).

To further characterize the effect of *VdMFS2* on pathogenicity of *V. dahliae*, Vd592, *VdMFS2* deletion mutants, and complementary strains were inoculated onto "Xinluzao 7" at the two-true-leaf stage using the root irrigation method. The cotton disease symptoms were examined at 14 and 28 days post inoculation (dpi). The results showed that the cotton seedlings inoculated with Δ*VdMFS2* mutants showed obvious leaf wilting, yellowing, and even shedding symptoms, while the cotton plants inoculated with Vd592

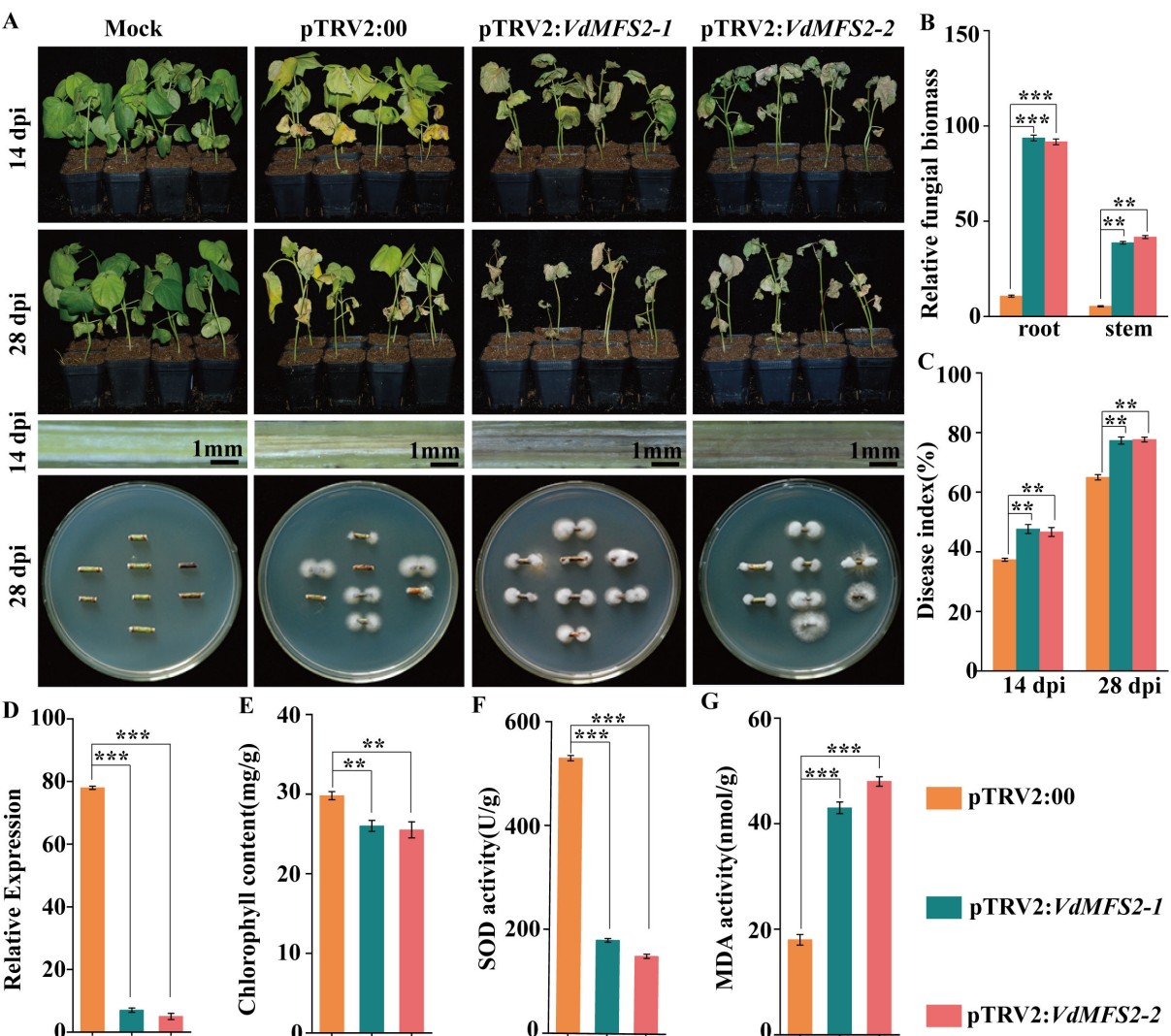

**FIG 2** The silencing of *VdMFS2* by HIGS resulted in increased susceptibility of cotton to VW. (A) Disease symptoms of HIGS-treated seedlings at 14 and 28 dpi. Vascular discoloration of stems from HIGS-treated cotton seedlings (Scale bar = 1 mm). Fungal hyphae were reisolated from the stem of HIGS-treated plants at 28 dpi and cultured for 5 days on PDA medium. (B) Quantification of fungal biomass in cotton roots and stems at 14 dpi. The qRT-PCR was used to determine the fungal biomass. (C) The disease index of HIGS-treated plants at 14 and 28 dpi. (D) Relative expression level of *VdMFS2* in stems of HIGS-treated plants at 14 dpi. The qRT-PCR was used to determine the relative expression of *VdMFS2*. (E) Chlorophyll content in leaves of HIGS treated plants at 14 dpi. (F) The activity of SOD enzyme in leaves of HIGS treated plants at 14 dpi. (G) MDA content in leaves of HIGS treated plants at 14 dpi. Data analysis was performed using R software. Significant differences between groups were determined by Student's *t*-test with a significance level of "**" for $P < 0.01$ and "***" for $P < 0.001$. Error bars represent standard deviation (SD).

and Δ*VdMFS2-C* strains only showed milder disease symptoms. Compared with the cotton plants inoculated with WT and Δ*VdMFS2-C* strains, the vascular bundles of cotton plants inoculated with Δ*VdMFS2* strains showed more severe browning, with a larger amount of isolated *V. dahliae* and a higher fungal biomass (Fig. 4B and D). At 14 and 28 dpi, the disease index of cotton plants infected by Δ*VdMFS2* mutants was significantly higher than that of WT and Δ*VdMFS2-C* strains (Fig. 4C). These results indicated that the *VdMFS2* deletion led to an increase in the pathogenicity of *V. dahliae*.

## *VdMFS2* deletion led to increased carbon sources utilization ability of *V. dahliae*

The increased growth of Δ*VdMFS2* mutants may be due to their ability to obtain more nutrients. To analyze their carbon-source utilization abilities, different strains were

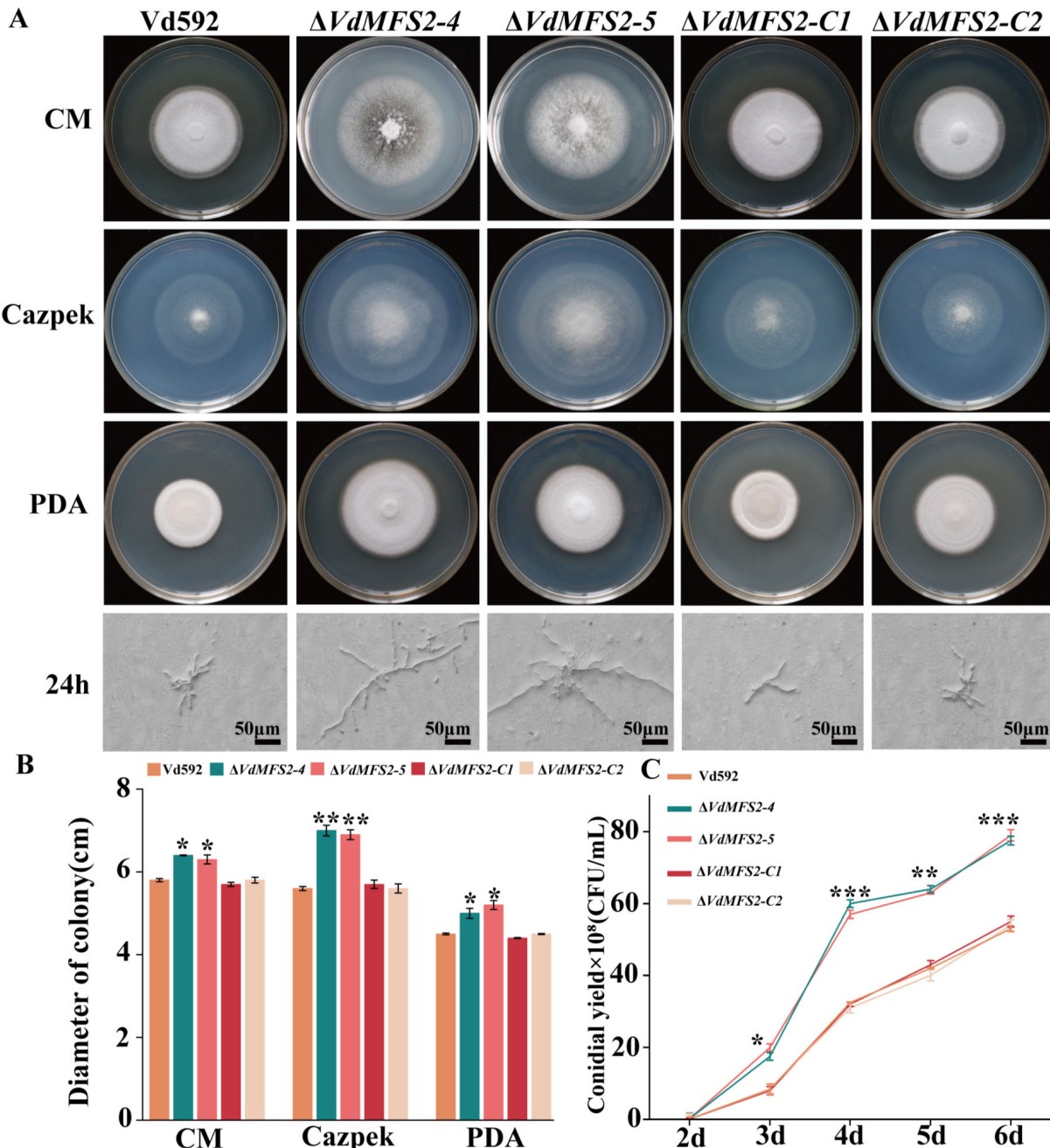

**FIG 3** *VdMFS2* deletion accelerated the growth of *V. dahliae*. (A) Colony morphology of all strains cultured on CM, Czapek-Dox medium, and PDA media at 20 days post-inoculation. Hyphal morphology observed on PDA medium at 24 h post-inoculation under a stereomicroscope. Scale bars: 50 μm. (B) Colony diameters of strains grown on CM, Czapek-Dox medium, and PDA media at 20 days post-inoculation. (C) Conidial production of *V. dahliae* strains in Czapek-Dox medium at 25°C. Conidia were quantified at 2–6 days post-inoculation. Data analysis was performed using R software. Significant differences between groups were determined by Student's *t*-test with a significance level of "*" for $P < 0.05$, "**" for $P < 0.01$, and "***" for $P < 0.001$. Error bars represent standard deviation (SD).

inoculated on Czapek-Dox medium supplemented with glucose, sucrose, xylose, pectin, and cellulose. The results showed that the colony diameters of the *VdMFS2* deletion mutants were obviously larger than those of the WT and complementary strains on media containing each carbon source, suggesting that *VdMFS2* deletion mutants exhibit enhanced carbon-source utilization (Fig. 5A and B).

The increased carbon-source utilization in Δ*VdMFS2* mutants led us to speculate that *VdMFS2* may not be involved in sugar transport. To test this hypothesis, the gene was

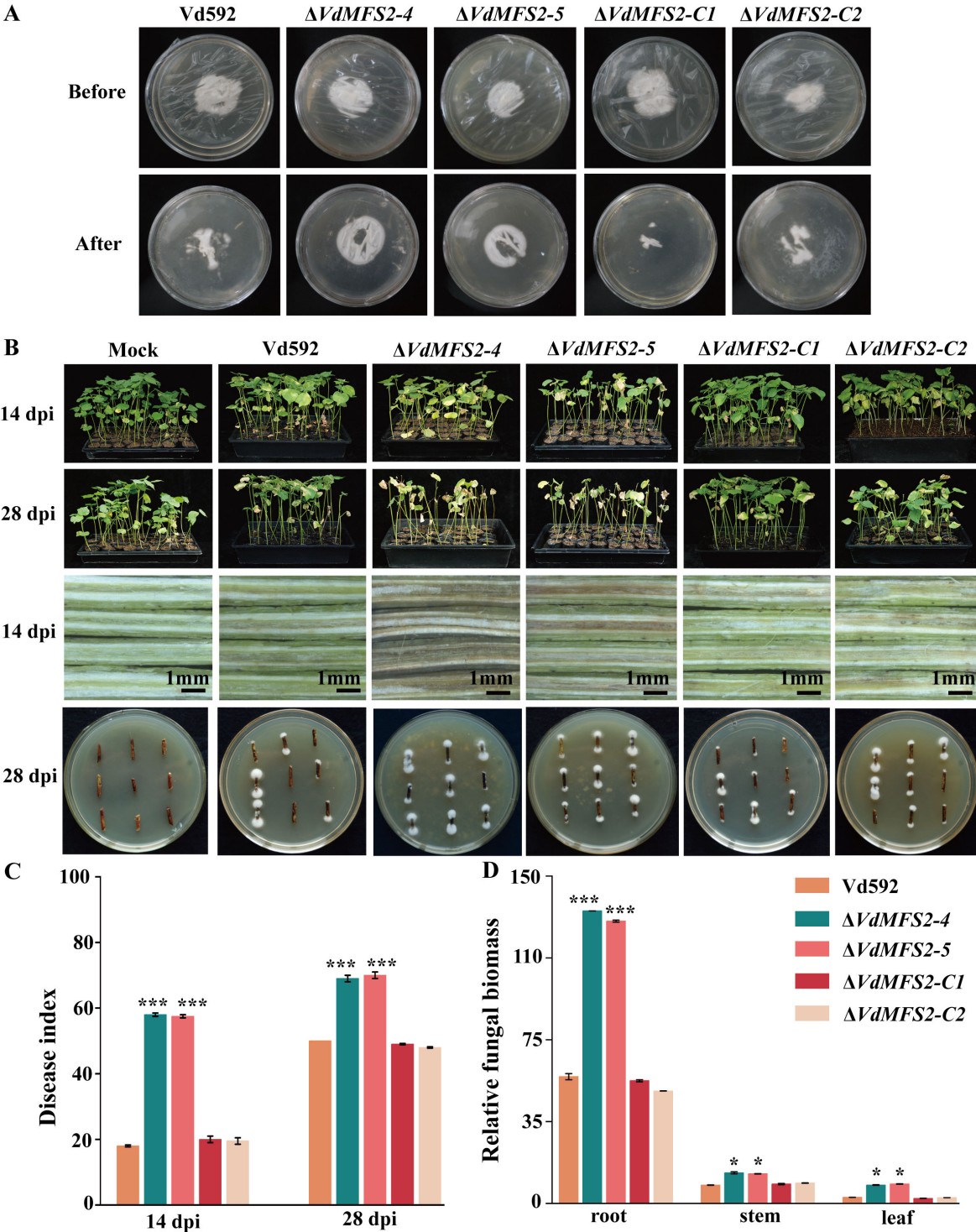

**FIG 4** *VdMFS2* deletion enhances pathogenicity of *V. dahliae*. (A) Different strains were cultured on cellophane-overlaid PDA medium for 7 days (Before), followed by cellophane removal and additional 3 days of incubation (After). (B) Disease symptoms in cotton plants inoculated with different strains at 14 and 28 dpi. Vascular discoloration in stems is shown (Scale bar = 1 mm). For fungal reisolation, the infected stems at 28 dpi were cut and then cultured on PDA medium for 5 days. (C) Disease index (DI) of cotton plants infected with different strains at 14 and 28 dpi. (D) Fungal DNA quantification in cotton roots, stems, and leaves at 28 dpi. Fungal DNA was detected using primers *Ve-ITS1*-F/ *ST-VE1*-R, with *GhUBQ7* as the endogenous control. Data analysis was performed using R software. Significant differences between groups were determined by Student's *t*-test with a significance level of "*" for $P < 0.05$ and "***" for $P < 0.001$. Error bars represent standard deviation (SD).

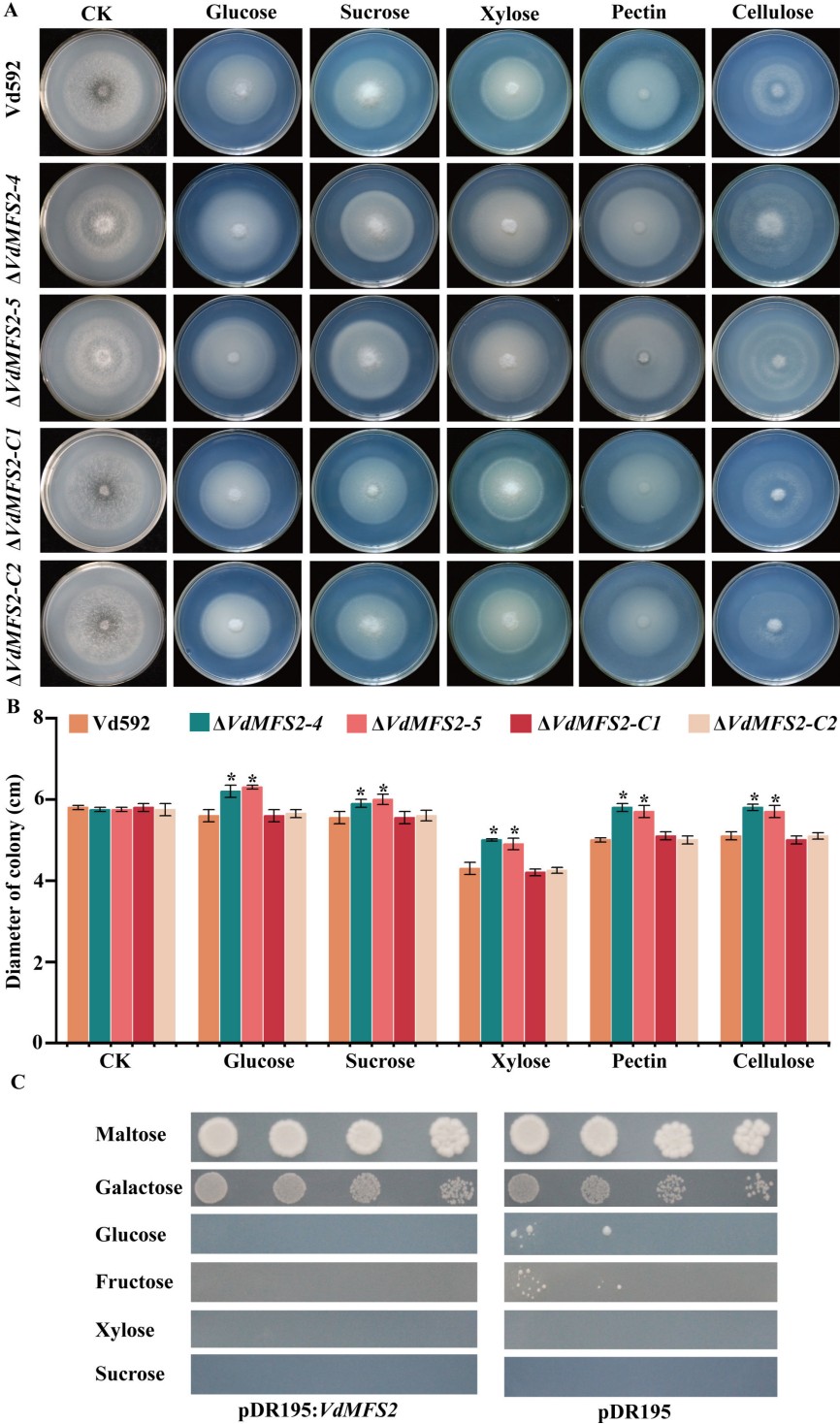

**FIG 5** *VdMFS2* deletion led to increased carbon sources utilization ability of *V. dahliae*. (A) Phenotypes of different strains on Czapek-Dox media supplemented with glucose, sucrose, xylose, pectin, or cellulose after 15 days of inoculation. (B) Colony diameters of all strains on Czapek-Dox media containing different carbon sources after 15 days of inoculation. (C) Growth of yeast mutant strain EBY.VW4000 carrying pDR195-*VdMFS2* on media containing different sugars. Yeast cell concentrations were $10^7$, $10^6$, $10^5$, and $10^4$ cells/mL (from left to right). Data analysis was performed using R software. Significant differences between groups were determined by Student's *t*-test with a significance level of "*" for $P < 0.05$. Error bars represent standard deviation (SD).

transformed into a yeast mutant strain EBY.VW4000 lacking sucrose and hexose transporters to evaluate its sugar-transport activity. The results showed that the yeast transformed with *VdMFS2* could not grow on Czapek-Dox media containing any of the sugar sources (glucose, fructose, xylose, and sucrose) (Fig. 5C). This result indicated that *VdMFS2* is not involved in the transportation of primary sugars, such as glucose, fructose, and sucrose.

## *VdMFS2* deletion led to reduced utilization ability of certain nitrogen sources of *V. dahliae*

Previous studies found that some members of the ACS subfamily are involved in the transport of amino acids, such as the oligopeptide/histidine transporters *PHT1* and *PHT2* (35) and vesicular glutamate transporter *VGLUT* (36). Therefore, the amino acids' utilization abilities of different strains were analyzed by inoculating them on Czapek-Dox medium supplemented with Phenylalanine (Phe), Leucine (Leu), Valine (Val), Cysteine (Cys), and Tryptophan (Trp), respectively. The results showed that the colony diameters of the ΔVdMFS2 mutants were obviously smaller than that of Vd592 and complementary strains on media containing Leu and Val, while larger on media containing Phe and Trp, and the same as that of Vd592 and complementary strains on media containing Cys (Fig. 6A and B). These results suggested that *VdMFS2* is involved in the transportation of Leu and Val in *V. dahliae*.

## *VdMFS2* deletion led to enhanced stress tolerance of *V. dahliae*

To assess the role of *VdMFS2* in stress sensitivity, different strains were cultured on PDA medium amended with $CuSO_4$, Congo red, NaCl, sodium dodecyl sulfate (SDS), or Calcofluor White (CFW). The *VdMFS2* deletion mutants exhibited lower growth inhibition rates than WT and complementary strains under these stress conditions (Fig. 7A and B).

Previous reports have shown that MFS transporters are involved in antifungal drug resistance (36). Therefore, ΔVdMFS2 mutants were tested on media containing the broad-spectrum fungicide carbendazim. At all concentrations tested (0.5 to 10 g/L), the inhibition zones around ΔVdMFS2 mutant colonies were significantly smaller than those of the wild-type and complementary strains (Fig. 8A), and the mutants exhibited lower growth-inhibition rates (Fig. 8B). These results indicate that deletion of *VdMFS2* enhances the tolerance of *V. dahliae* to this fungicide.

## *VdMFS2* deletion led to an enhancement in protein secretion of *V. dahliae*

Secreted proteins play important roles in the pathogenicity of pathogens (37, 38). The increased pathogenicity of ΔVdMFS2 mutants may be caused by changes in the expression of secreted proteins. Therefore, the secreted proteins of ΔVdMFS2 mutants and WT were collected and quantified after cultivated in Czapek-Dox media for 5 days. The secreted proteins were then extracted by $(NH_4)_2SO_4$ precipitation, and the content was calculated using a bovine serum albumin (BSA) standard curve. It was found that the concentration of secreted proteins from ΔVdMFS2 mutants was significantly higher than that of WT and ΔVdMFS2-C strains (Fig. 9A). SDS-PAGE electrophoresis showed that the secreted proteins of two ΔVdMFS2 mutants had more and clearer bands than WT and ΔVdMFS2-C strains under the same volume (Fig. 9B). After injecting the same concentration of secreted proteins from different strains, the cotton leaves injected with secreted proteins of ΔVdMFS2 mutants had a larger DAB stained area, suggesting that the secreted proteins of ΔVdMFS2 mutants can induce the production of more $H_2O_2$ compared with WT and complementary strains (Fig. 9C and D). These results indicated that ΔVdMFS2 mutants produce more secreted proteins than WT and complementary strains.

To further investigate whether ΔVdMFS2 was capable of producing more secreted proteins, comparative proteomic analysis was carried out on the exoproteome of WT and ΔVdMFS2-4 mutant. A total of 63 differentially expressed proteins (DEPs) were identified. Some intracellular proteins may be present in exoproteome proteins due to cellular lysis

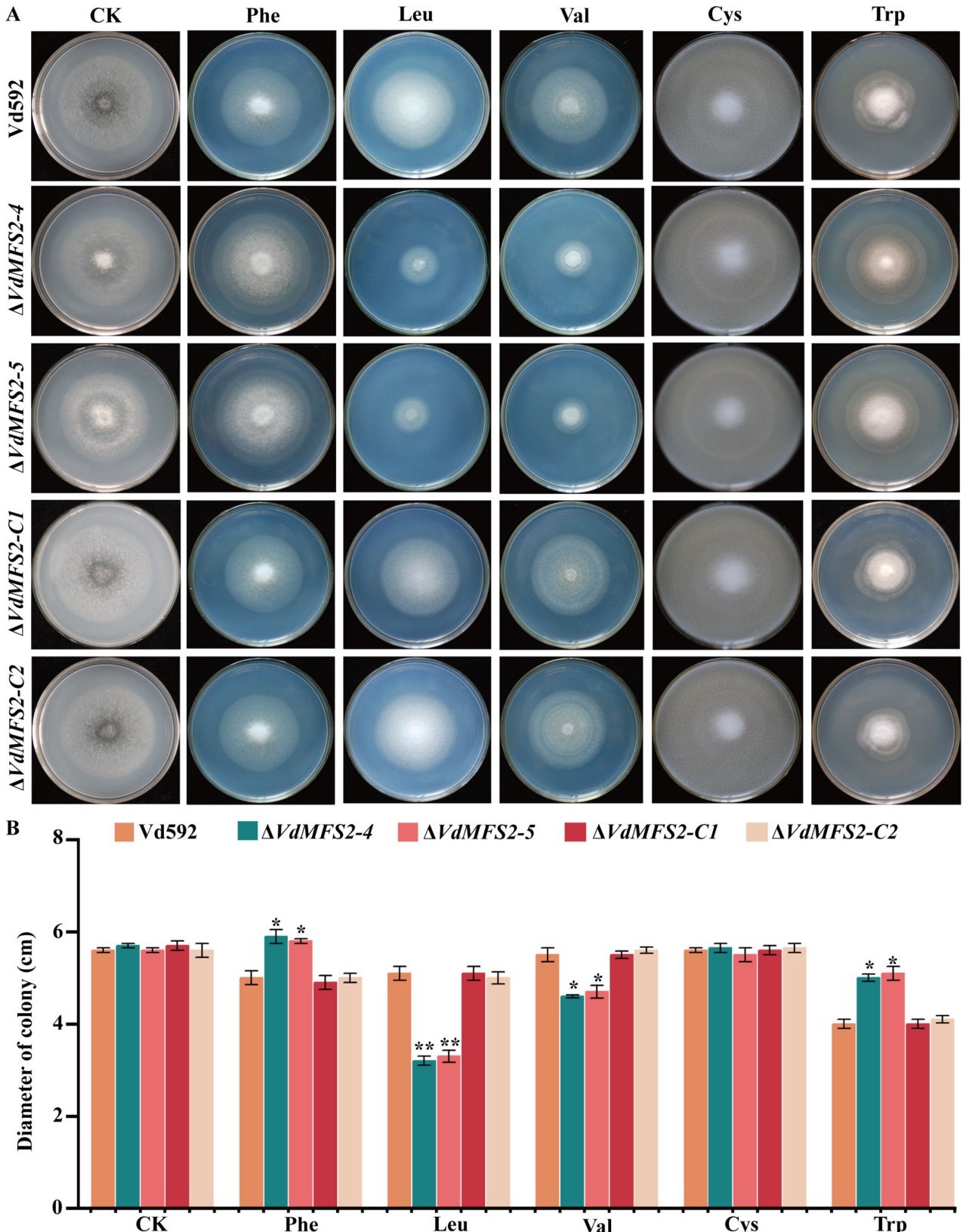

**FIG 6** *VdMFS2* deletion led to reduced utilization ability of certain nitrogen sources of *V. dahliae*. (A) Phenotypes of the different strains on Czapek-Dox supplemented with Phe, Leu, Val, Cys, or Trp after 15 days of inoculation. (B) Colony diameters of all strains on media containing different nitrogen sources after 15 days of inoculation. Data analysis was performed using R software. Significant differences between groups were determined by Student's *t*-test with a significance level of "*" for $P < 0.05$ and "**" for $P < 0.01$. Error bars represent standard deviation (SD).

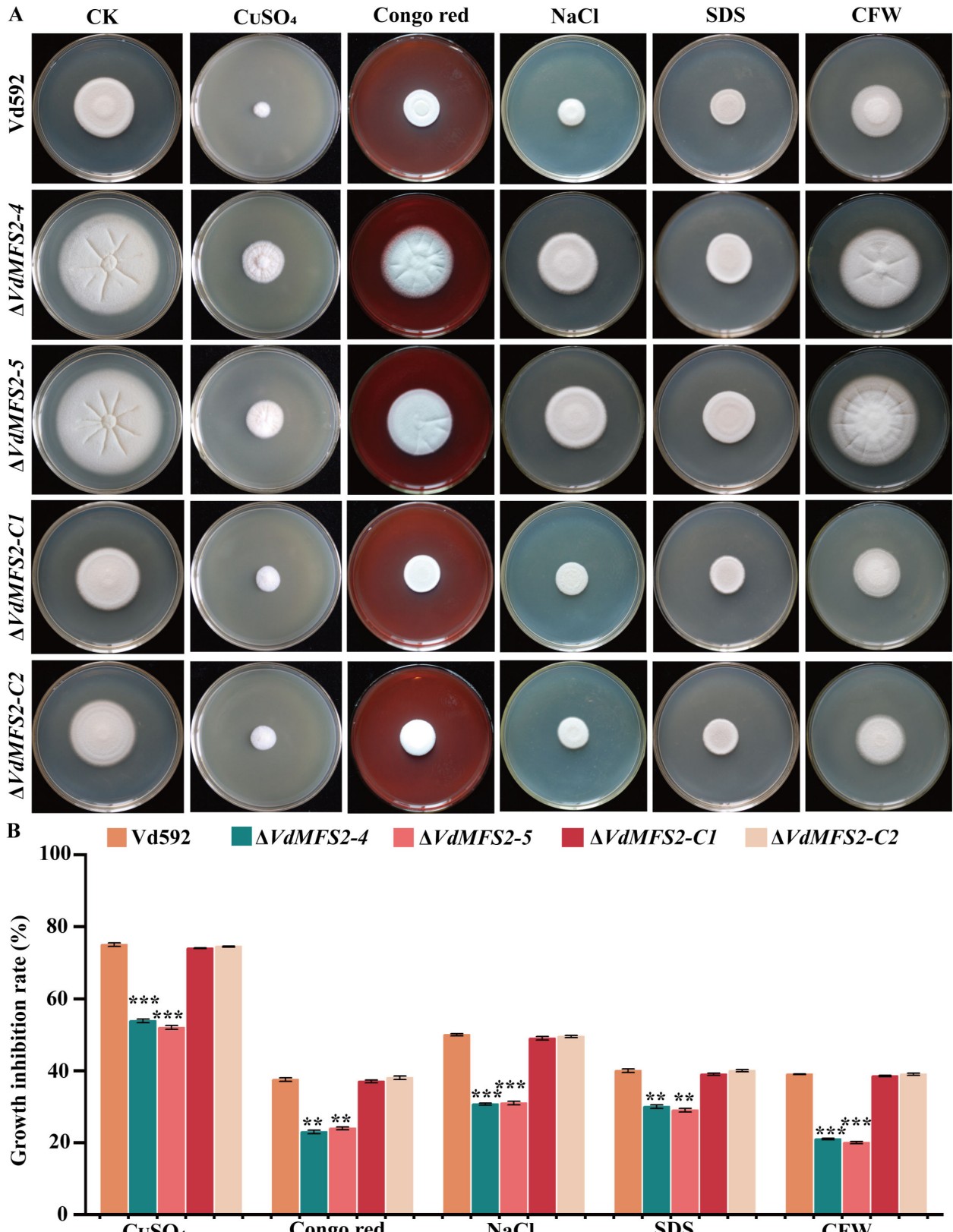

**FIG 7** *VdMFS2* deletion enhanced tolerance of *V. dahliae* to various stresses. (A) Different strains on PDA media supplemented with $CuSO_4$, Congo red (CR), NaCl, SDS, or CFW. (B) Growth inhibition rates of all strains under different stresses. Data analysis was performed using R software. Significant differences between groups were determined by Student's *t*-test with a significance level of "**" for $P < 0.01$ and "***" for $P < 0.001$. Error bars represent standard deviation (SD).

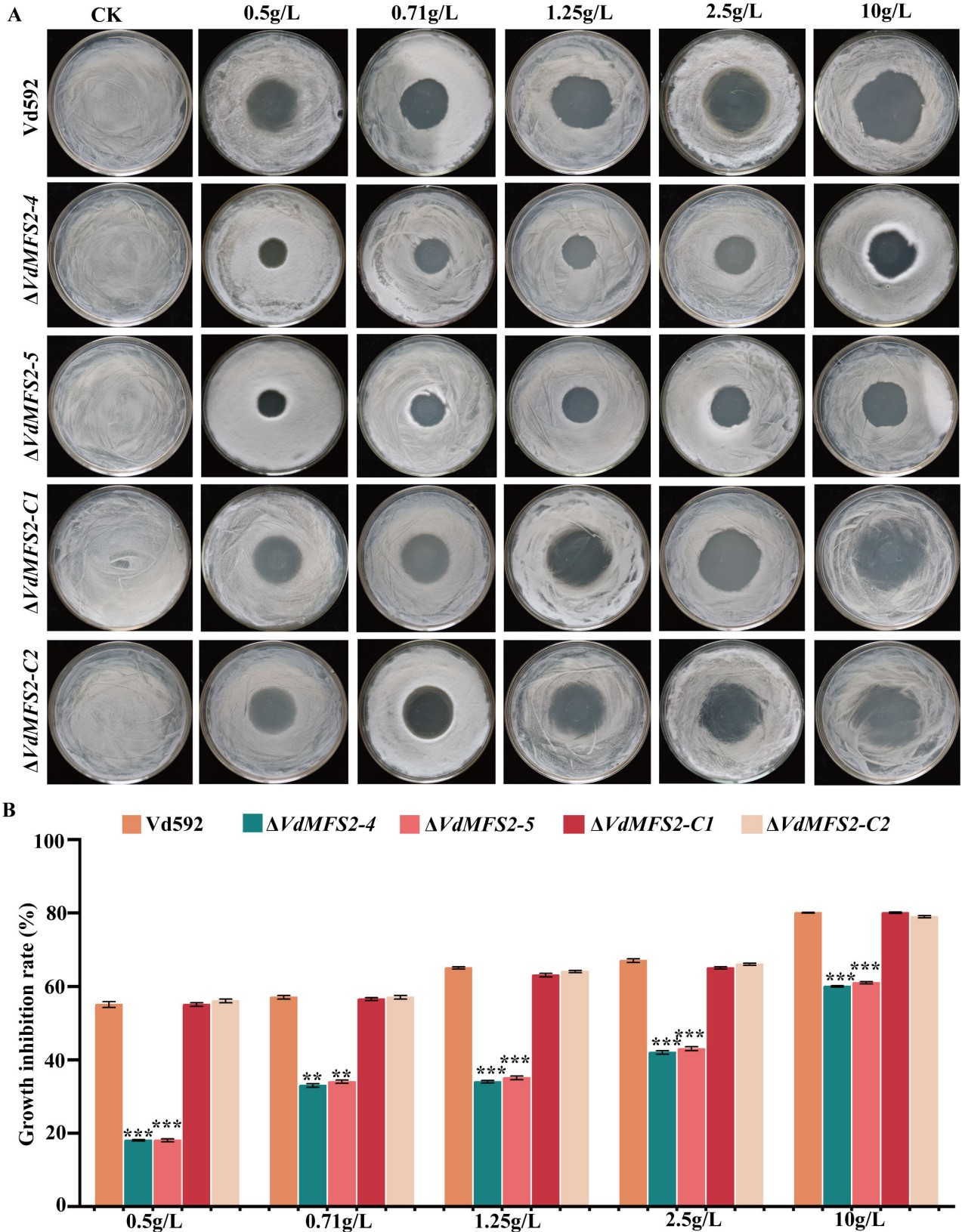

**FIG 8** *VdMFS2* deletion enhanced tolerance of *V. dahliae* to various stresses. (A) Different strains on PDA media supplemented with carbendazim. (B) Growth inhibition rates of all strains on PDA media containing different concentrations of carbendazim. Data analysis was performed using R software. Significant differences between groups were determined by Student's *t*-test with a significance level of "**" for $P < 0.01$ and "***" for $P < 0.001$. Error bars represent standard deviation (SD).

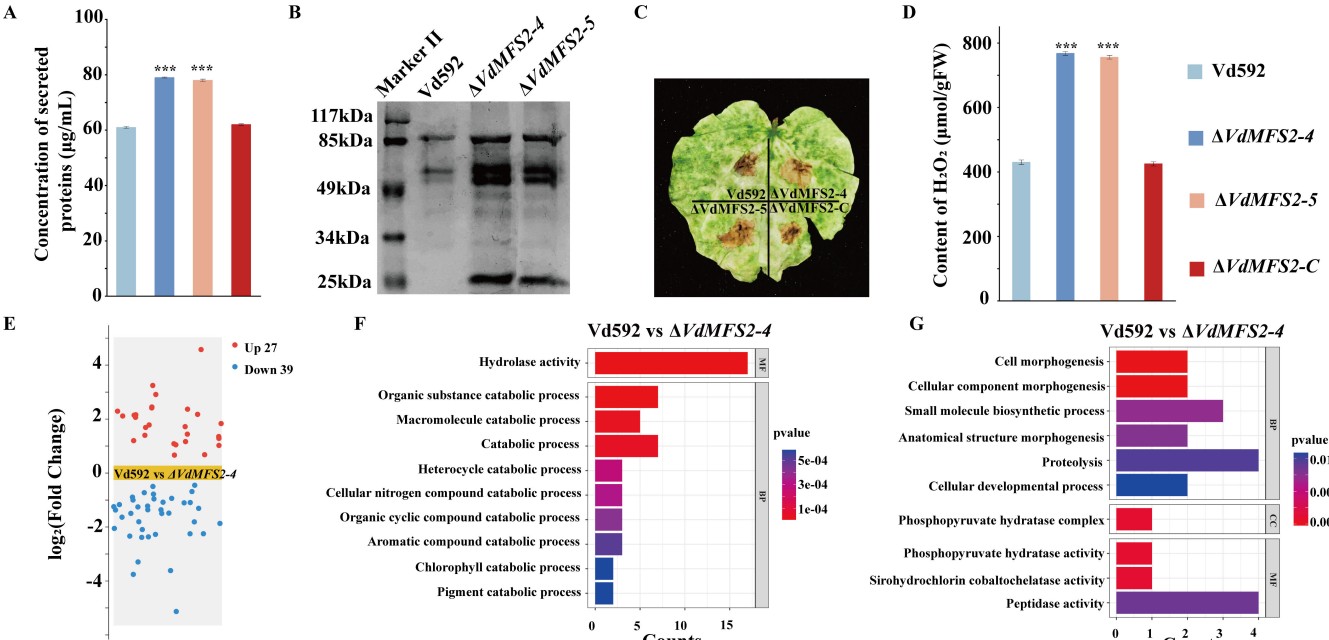

**FIG 9** *VdMFS2* deletion led to an increase in protein secretion of *V. dahliae*. (A) The concentration of secreted proteins from different strains. (B) The gel electrophoresis of secreted proteins from different strains in the same volume. (C) DAB stained area on cotton leaves after injecting secreted proteins from different strains for 3 days. (D) The content of $H_2O_2$ produced on cotton leaves after injecting secreted proteins from different strains for 3 days. (E) Volcanic plots of DEPs in Δ*VdMFS2-4* vs Vd592 comparison. Red and blue points represent the upregulated and downregulated DEPs, respectively. (F and G) GO enrichment analysis of the up- and downregulated DEPs in Δ*VdMFS2-4* vs Vd592 comparison. X-axis represents the DEPs' counts, and Y-axis represents the top 10 enriched GO terms, different colors represent the p-value. Data analysis was performed using R software. Significant differences between groups were determined by Student's *t*-test with a significance level of "***" for *P* < 0.001. Error bars represent standard deviation (SD).

during the culture process. Therefore, secreted proteins with signal peptide, but lacking transmembrane domains were further identified by using SignalP 4.1 and TMHMM 2.0, and secreted proteins without signal peptide were identified by using SecretomeP. After removing intracellular proteins, a total of 51 differentially expressed proteins (DEPs) were identified, including 32 upregulated and 19 downregulated proteins in Δ*VdMFS2-4* mutant. More upregulated secreted proteins were found in the Δ*VdMFS2-4* mutant, which was consistent with the protein concentration determination (Fig. 9E) and gel electrophoresis results (Fig. 9B). GO enrichment analysis showed that the upregulated DEPs were mainly enriched in hydrolase activity term, and the downregulated ones mainly enriched in peptidase activity term (Fig. 9F and G).

## Comparative proteomic analysis of *V. dahliae*-cotton interaction

To compare the interaction differences between *V. dahliae* strains in the early infection response of cotton, the WT and Δ*VdMFS2-4* strains were co-cultured with root tissues from Xinluzao seven in Czapek-Dox for 5 days, respectively, which were conducted simultaneously with the cultivation of two strains in Czapek-Dox media. The samples cultivated in Czapek-Dox media were designated as Vd592 and Δ*VdMFS2-4*, and the samples co-cultured with root tissues as Vd592Y and Δ*VdMFS2-4*Y (Fig. 10). The secreted proteins were precipitated and collected for proteomic sequencing.

After removing intracellular proteins, a total of 29 DEPs (seven upregulated and 22 downregulated) were identified in Vd592Y vs Vd592 comparison (Fig. 11A), 71 DEPs (55 upregulated and 16 downregulated) in the Δ*VdMFS2-4*Y vs Δ*VdMFS2-4* comparison (Fig. 11D), and 48 DEPs (47 upregulated and 1 downregulated) in the Δ*VdMFS2-4*Y vs Vd592Y comparison (Fig. 11G). The results suggested that the Δ*VdMFS2-4* mutant produced more secreted proteins during the interaction with cotton compared to Vd592. GO enrichment analysis was conducted for the upregulated and downregulated DEPs

in each comparison (Fig. 11B, C, E, F, H and I). It was notable that up-regulated DEPs in Δ*VdMFS2-4*Y vs Δ*VdMFS2-4* and Δ*VdMFS2-4*Y vs Vd592Y comparisons were mainly enriched in hydrolase activity term.

## Pathogenicity-related proteins of *V. dahliae*-cotton interaction

Since the Δ*VdMFS2-4* mutant produces more secreted proteins before and after induction of cotton root, the upregulated DEPs in Δ*VdMFS2-4* vs Vd592, Δ*VdMFS2-4*Y vs Δ*VdMFS2-4,* and Δ*VdMFS2-4*Y vs Vd592Y comparisons were believed to be responsible for the enhanced pathogenicity of the *VdMFS* deletion mutants. A Venn diagram showed that there were 23, 24, and 9 secreted proteins unique to the Δ*VdMFS2-4* vs Vd592, Δ*VdMFS2-4*Y vs Δ*VdMFS2-4*, and Δ*VdMFS2-4*Y vs Vd592Y comparisons, respectively (Fig. 12A).

A total of 94 secreted proteins (SPs) were identified, including 55 classical secreted proteins (CSPs) and 40 non-classical secreted proteins (N-CSPs) (Fig. 12B). There were 23 carbohydrate active enzymes (CAZymes) and 18 small cysteine-rich proteins (SCRPs, <400 amino acids, ≥4 cysteine residues) identified from 55 CSPs, and 14 pathogen-host interaction proteins (PHIs) included in 94 SPs (Fig. 12B) (Table S2). Among 23 CAZymes, 12 were glycoside hydrolases (GHs), 4 auxiliary activity (AAs), 4 carbohydrate esterases (CEs), 2 carbohydrate-binding modules (CBMs), and 1 polysaccharide lyase (PL) (Fig. 12C), and the expression levels of these proteins were upregulated in at least one comparison (Fig. 12D). Several cell-wall degrading enzymes (CWDEs) were found in these proteins, including five cellulose (G2WV74, G2WQ86, G2WZZ8, G2WQ69, and G2WX35) and four hemicellulose (G2WUU1, G2WW80, G2WZT4, and G2WWW4). Based on this result, the DNS method was used to measure the activities of cellulase, hemicellulase, and pectinase. It was found that the enzyme activities degrading cellulose, xylan, and pectin in the Δ*VdMFS2* strains were significantly higher than those in the Vd592 and complementary strains (Fig. 12E through G), further suggesting that Δ*VdMFS2* produces more cell-wall degrading enzymes compared with wild type. Among the cell-wall-degrading enzymes, we selected the hemicellulase G2WZT4 (α-glucuronidase; gene ID VDAG_03526), designated it *VdAgl1*, and performed HIGS assays. Silencing *VdAgl1* significantly attenuated the pathogenicity of *V. dahliae* (Fig. S1A).

Previous reports found that SCRPs and N-CSPs play important roles in fungal pathogenicity (39). The expression levels of 18 SCRPs and 40 N-CSPs were identified. Among the 18 small cysteine-rich proteins (SCRPs), several differentially expressed proteins are related to hydrolase classes, including β-glucosidase (G2WYQ1) and cerato-platanin (G2WWQ6); binding proteins, including ubiquitin three binding protein (G2WTI6) and tyrosinase copper-binding protein (G2X9I2); cytochrome and redox proteins, including cytochrome b-c1 (G2WR07) and ubiquinol-cytochrome c (G2WSE5). Among them, two SCRPs have been previously reported: G2WWQ6, a member of the classic fungal effector

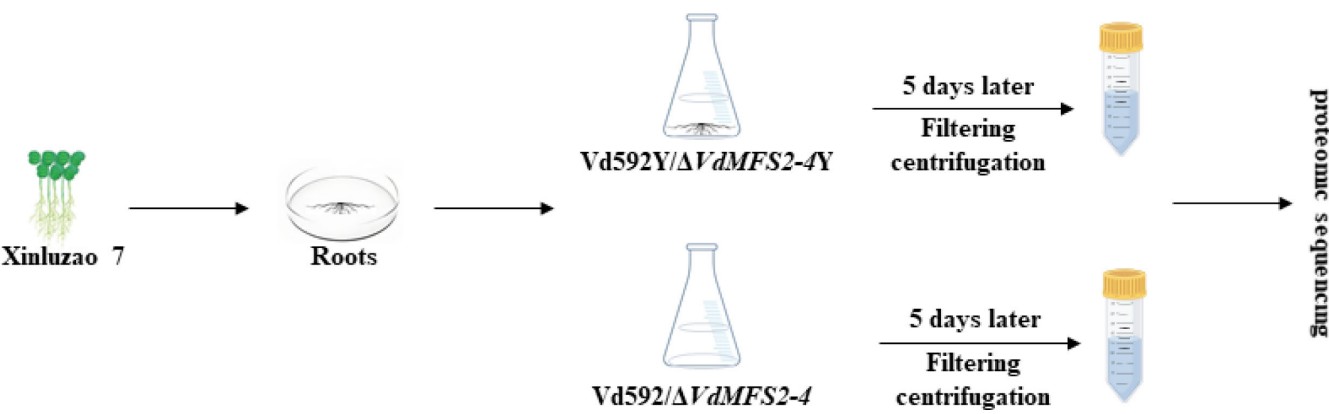

**FIG 10** The sampling mode diagram of the proteome.

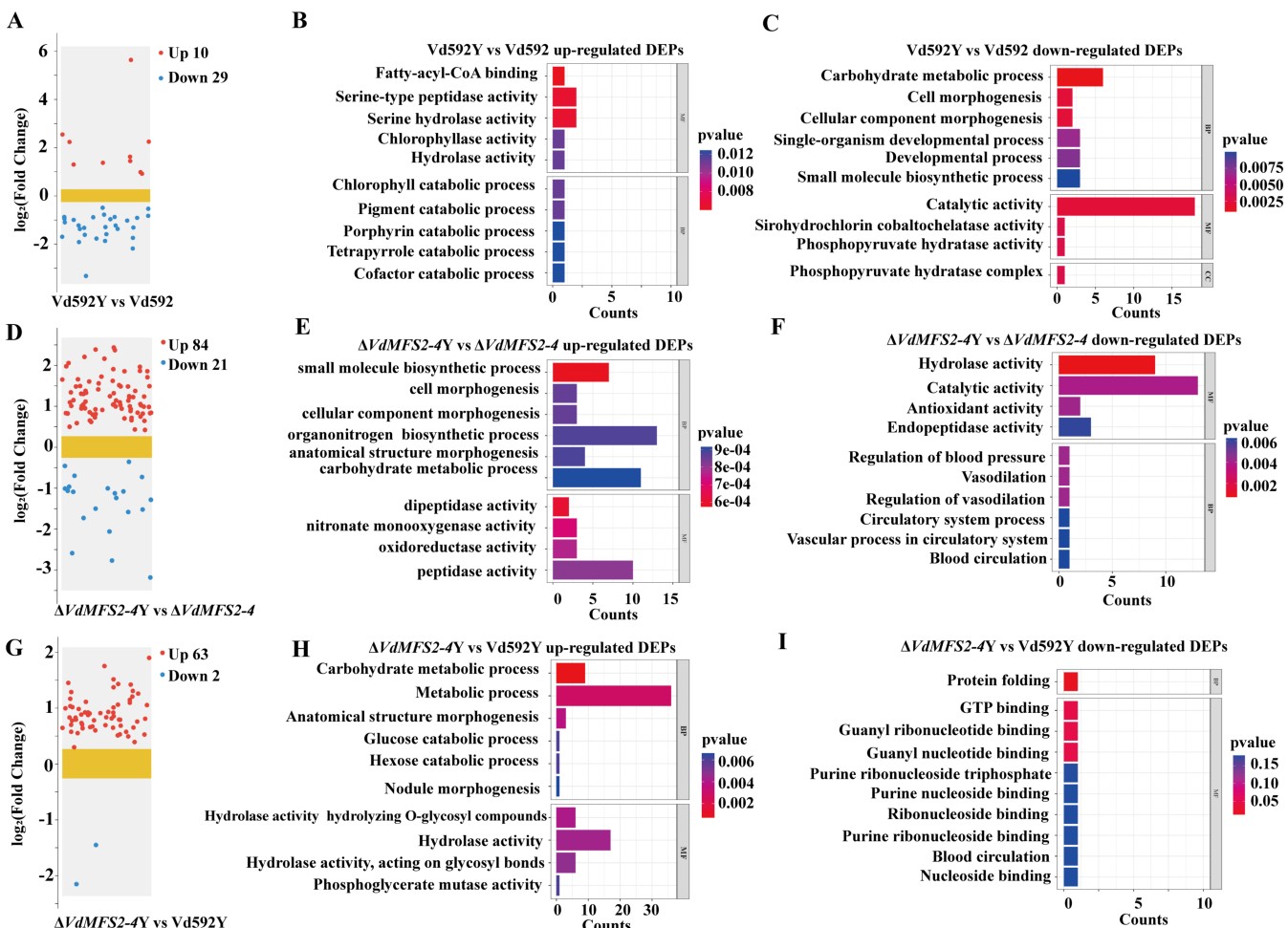

**FIG 11** Comparative proteomic analysis of *V. dahliae*-cotton interaction. (A) Volcanic plots of DEPs in Vd592Y vs Vd592 comparison. (B and C) GO enrichment analysis of the up- and downregulated DEPs in Vd592Y vs Vd592. (D) Volcanic plots of DEPs in Δ*VdMFS2-4*Y vs Δ*VdMFS2-4* comparison. (E and F) GO enrichment analysis of the up- and downregulated DEPs in Δ*VdMFS2-4*Y vs Δ*VdMFS2-4* comparison. (G) Volcanic plots of DEPs in Vd592Y vs Δ*VdMFS2-4*Y comparison. (H and I) GO enrichment analysis of the up- and downregulated DEPs in Vd592Y vs Δ*VdMFS2-4*Y. Red and blue points represent the upregulated and downregulated DEPs, respectively. X-axis represents the DEPs' counts, and Y-axis represents the top 10 enriched GO terms, *P* < 0.05, FC ≥ 1.2.

family (*VdCP1*) that can trigger plant immunity, and G2WYQ1, a secreted glycosidase (*Vdβglu1*) involved in cell-wall degradation and virulence enhancement through ER-stress induction (Fig. S1B) (40, 41). Among 40 N-CSPs, several differentially expressed proteins were associated with key biological processes. Metabolic-related proteins included β-glucosidase (G2WTZ0, G2WYQ1), endoglucanase-1 (G2WWA4), and 6-phosphogluconate dehydrogenase (G2X7Q5). Protein synthesis-related factors comprised multiple ribosomal subunits, such as 60S ribosomal protein L22 (G2WRJ5), L23a (G2X748), and L3 (G2XAE8), as well as 40S ribosomal protein S0 (G2XAI6) and S6 (G2XAQ5). Additionally, antioxidant-related proteins were identified, including catalase-peroxidase (G2WX55) and glutathione transferase (G2WUU6) (Fig. S1C), and their functions in pathogenicity of *V. dahliae* need further research.

## DISCUSSION

Previous studies have shown that MFS transporter genes are involved in the interaction between host plants and fungi, can uptake host sugars. For example, members *NcHXT-1* and *NcHXT-2* of the MFS family in the lignocellulose-degrading fungus *Neurospora crassa* are crucial for the transport of glucose, galactose, or mannose (42). In wild-type

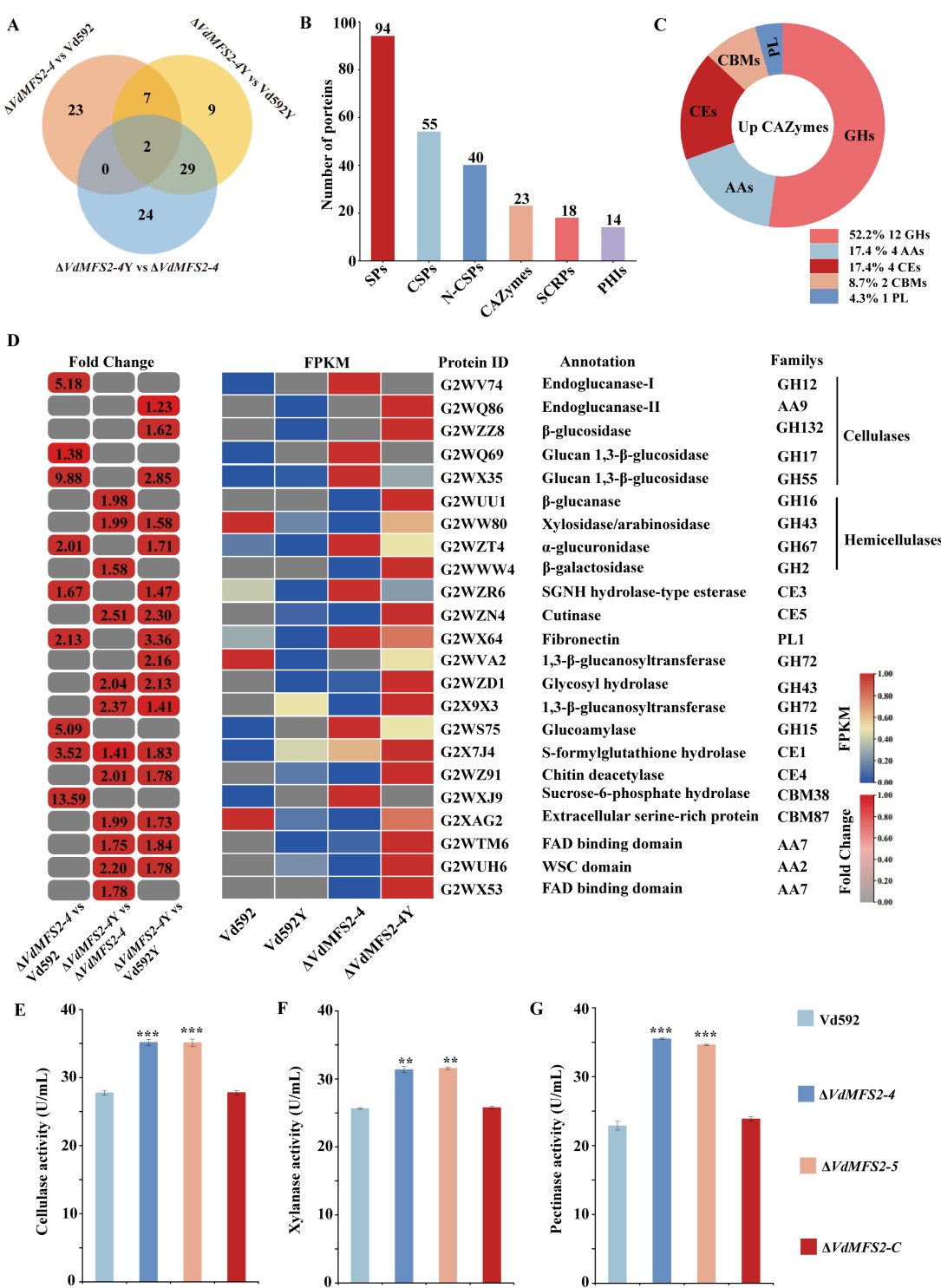

**FIG 12** Pathogenicity-related DEPs. (A) Venn diagram of upregulated DEPs in Δ*VdMFS2-4* vs Vd592, Δ*VdMFS2-4*Y vs Δ*VdMFS2-4*, and Δ*VdMFS2-4*Y vs Vd592Y comparisons. (B) The number of DEPs encoding secreted proteins (SPs), classical secreted proteins (CSPs), non-classical secreted proteins (N-CSPs), carbohydrate-active enzymes (CAZymes), small cysteine-rich proteins (SCRPs), and pathogen-host interaction-related proteins (PHIs). (C) Pie graph showing the number of upregulated CAZymes in Δ*VdMFS2-4* vs Vd592, Δ*VdMFS2-4*Y vs Δ*VdMFS2-4*, and Δ*VdMFS2-4*Y vs Vd592Y comparisons. (D) Heatmap showing the expression fold change of CAZymes in different comparisons. (E–G) The activities of cellulase, xylanase, and pectinase of different strains. The heatmap was generated based on the proteomics data. The numbers in the heatmap represent fold change, $P < 0.05$, FC ≥ 1.2. Data analysis was performed using R software. Significant differences between groups were determined by Student's $t$-test with a significance level of "**" for $P < 0.01$ and "***" for $P < 0.001$. Error bars represent standard deviation (SD).

*Saccharomyces cerevisiae*, Xltr1p, another member of the MFS family, significantly enhances the uptake of D-xylose when overexpressed (33, 43). The MFS family is mainly divided into 17 subfamilies (19). Through evolutionary relationships, it has been found that *VdMFS2* belongs to the ACS subfamily in this study. It has been reported that members of the ACS family can transport amino acids (36). The oligopeptide/histidine transporters PHT1 and PHT2 are two proteins of the ACS subfamily. They mediate the transmembrane transport of histidine and certain dipeptides and tripeptides via a proton gradient (44). The vesicular glutamate transporter *VGLUT*, another member of the ACS family, is responsible for loading glutamate into synaptic vesicles (37). In the present study, we observed that the deletion of *VdMFS2* resulted in a pronounced inhibition of colony growth for the ΔVdMFS2 mutant strains in media containing leucine (Leu) and valine (Val) as the sole nitrogen sources. Additionally, colony growth of ΔVdMFS2 mutant strains was enhanced in media with varying carbon sources, and a yeast mutant strain EBY.VW4000 lacking sucrose and hexose transporters transformed with *VdMFS2* could not grow on Czapek-Dox media containing any of the sugar sources (glucose, fructose, xylose, and sucrose). This result indicated that *VdMFS2* is involved in the transport of Leu and Val, while not being involved in the transportation of primary sugars, such as glucose, fructose, and sucrose.

In phytopathogenic fungi, MFS transporters can pump mycotoxins, thereby enhancing the fungal virulence towards host plants (45). Additionally, the overexpression of MFS transporters has been associated with multidrug resistance (MDR) to a variety of chemical drugs, endowing fungi with a multidrug-resistant phenotype (36). Putative MFS-MDR transporters are widely distributed among pathogenic fungi such as *Candida*, *Cryptococcus*, and *Aspergillus* (36, 46). In the genome of *S. cerevisiae*, a total of 22 putative MFS drug transporters have been reported (22). However, the role of MFS transporters in MDR remains not well understood. In other fungi, an MFS transporter gene *Flr1p* contributes to resistance against antifungal agents fluconazole and cycloheximide (47). Another MFS transporter gene *Pdr5p* mediates tolerance to these two compounds and is clearly a major determinant in the resistance phenotype to these drugs (37). In *Candida albicans*, two MFS transporters, *CaMDR1* and *Flu1*, are involved in azole and fluconazole resistance, respectively, but the native functions of these transporters are unknown, likely due to a lack of characterization efforts (38, 48). Contrary to previous research, the *VdMFS2* deletion mutant exhibited extremely high tolerance to carbendazim in this study, suggesting the gene has a negative regulatory effect on stress resistance of *V. dahliae* (Fig. 7 and 8).

Fungi secrete an array of carbohydrate-active enzymes (CAZymes) during host invasion to degrade polysaccharides, such as plant cell walls, which facilitates pathogen infection and nutrient acquisition from plant cells (49). For example, *Fusarium oxysporum* secretes a glycoside hydrolase, *FoEG1*, during the early stages of plant invasion, which aids in pathogen entry (50). CAZymes are primarily classified into six categories based on their catalytic modules or functional domains: carbohydrate esterases (CEs), glycoside hydrolases (GHs), polysaccharide lyases (PLs), glycosyltransferases (GTs), auxiliary activities (AAs), and carbohydrate-binding modules (CBMs) (51). Studies have revealed that the enzymatic activities of different fungi exhibit preferences for various plant types and adapt to their lifestyle (52, 53). Biotrophic pathogens, which are adapted to a biotrophic lifestyle within their hosts, typically possess a reduced repertoire of CAZymes (54). In contrast, hemibiotrophic and necrotrophic pathogens exhibit a significant increase in the families of glycoside hydrolases and polysaccharide lyases compared with biotrophic pathogens (55). Parasitic fungi, which are highly active in degrading plant material, contain a greater abundance of CAZymes (56). In this study, 23 upregulated CAZymes were identified across the ΔVdMFS2-4 vs Vd592, ΔVdMFS2-4Y vs ΔVdMFS2-4, and ΔVdMFS2-4Y vs Vd592Y comparisons, providing evidence for the enhanced virulence associated with the ΔVdMFS2 mutant (Fig. 12C and D).

The plant cell wall, primarily composed of cellulose, hemicellulose (especially xylan), pectin, lignin, and a small amount of structural proteins, is a dynamic structure that

plays a crucial role in preventing pathogen invasion (57, 58). The cell-wall-degrading enzymes (CWDEs) produced by pathogens are essential for colonization in host plants (59). Genomic sequence analysis of *V. dahliae* reveals a large number of CWDEs, including pectinases, xylanases, cellulases, and proteases (60). Previous studies have demonstrated that the sucrose non-fermenting protein kinase gene *VdSNF1* and the specific secreted protein gene *VdSSP1* positively regulate the activity of CWDEs, which is critical for the pathogenicity of *V. dahliae* on host plants (61, 62). Additionally, the pathogenicity-related genes *VdPR1* and *VdPR3* influence the pathogenicity of *V. dahliae* by modulating cellulase activity. Polygalacturonase *VdPG1* and xylanase *VdXyn4* digest pectin and xylan in the cell wall, respectively, enhancing the pathogenicity of *V. dahliae* to cotton (63–66). The xylanase member *VdXyn4* is involved in the degradation process of xylan, pectin, and cellulose in host cells and can cause necrosis of vascular bundle cells, such as leaf veins and petioles (66). In this study, 12 CWDEs were identified in the comparisons of Δ*VdMFS2-4* vs Vd592, Δ*VdMFS2-4*Y vs Δ*VdMFS2-4*, and Δ*VdMFS2-4*Y vs Vd592Y (Fig. 12D), as reported for *FgMFS1*, that transporter regulates deoxynivalenol (DON) export, and its deletion reduces expression of pectinase (pel) and cellulase (cel) genes by more than 50%, impairing host cell-wall degradation. Following this rationale, we selected the hemicellulase G2WZT4 (α-glucuronidase) identified among the carbohydrate hydrolases, performed HIGS silencing, and found that it significantly attenuates the virulence of *V. dahliae* in (Fig. S1A). Further indicating that the genes specifically responding to the host in highly pathogenic *V. dahliae* may play significant roles in the virulence differentiation of *V. dahliae*.

SCRPs generally exhibit the characteristics of low molecular weight and cysteine richness (67). For instance, a set of cysteine-rich fungal extracellular membrane domain proteins in *Fusarium graminearum* interact with *ZmLRR5* and *ZmWAK17ET* proteins in maize, thereby compromising *ZmWAK17*-mediated resistance (68). The *Phytophthora sojae* effector *PsXEG1* utilizes a signal peptide to target the extracellular space and induce cell death (69). To date, a variety of effectors have been reported in *V. dahliae*, including the cellulose-binding family module *VdCBM1* (70), the small cysteine-rich protein (SCRP) VdSCP41 (71), the lysin motif protein *Vd2LysM* (72), and the deacetylase *VdPD1*. These proteins facilitate the successful infection of pathogens by modulating plant immune responses (73). In this study, 18 SCRPs were identified (Fig. S1B), including two previously reported examples: G2WWQ6 (*VdCP1*), a classic fungal effector that can trigger plant immunity, and G2WYQ1 (*Vdβglu1*), a secreted glycosidase involved in cell-wall degradation and virulence enhancement through ER-stress induction (40, 41). The identification of this expanded set of SCRPs further substantiates the enhanced pathogenicity associated with the Δ*VdMFS2* mutation.

Unconventional protein secretion (N-SCPs) is defined as the process by which eukaryotic cells export proteins that are incapable of entering the conventional endoplasmic reticulum (ER)-Golgi pathway (74). These proteins lack signal peptides and are secreted via pathways other than the ER-Golgi route and are thus termed unconventional secreted proteins. Their secretion pathways are collectively referred to as N-SCPs pathways (75). Unlike the classical secretion pathway, the unconventional secretion modes of regulatory proteins are diverse and depend on the type of secreted protein, the stimuli the cell receives, and the cell type (75). N-SCPs play a significant role in the infection process of pathogens (76). In this study, we identified a total of 40 N-SCPs (Fig. S1C), which are associated with functions, such as metabolism, protein synthesis, and antioxidant activity. Among them, *Tpk1* and *Tpk2* have been reported in *Candida auris* and are associated with its multidrug resistance (77). *TmGst* has been reported in *Candida albicans*, which utilize glutathione transferase to combat oxidative stress. Glutathione transferase aids fungi in coping with host immune responses by scavenging free radicals (78). β-Glucosidase was found in fungi such as *Candida albicans*, *Fusarium*, and *Saccharomyces cerevisiae*, assisting fungi in degrading host cell walls and promoting the infection process. Additionally, glycoside hydrolases facilitate the degradation of host

carbohydrate molecules, providing nutrients for fungal growth (79–81). Evidently, N-SCPs play a critical role in the process of invasion of *V. dahliae* to hosts.

## ACKNOWLEDGMENTS

This research was supported by the Key Project of the Natural Science Foundation of the Xinjiang Production and Construction Corps (No. 2024DA001), National Natural Science Foundation of China (No. 32160615), Xinjiang Tianshan Talents Program (No. SN-SHZU-202402), Natural Science Support Program Project of the Xinjiang Production and Construction Corps (No. 2024DA017), and Science and Technology Plan Project of Shihezi 2024ZD03, Shihezi University Breeding Special Project (No. YZZX202304).

## AUTHOR AFFILIATIONS

[1]The Key Laboratory of Oasis Eco-Agriculture, Agriculture College, Shihezi University, Shihezi, People's Republic of China
[2]Jun Ken Agricultural Research Institute, Shihezi, People's Republic of China
[3]College of Life Sciences and Medicine, Zhejiang Sci-Tech University, Hangzhou, People's Republic of China

## AUTHOR ORCIDs

Yuqiang Sun ⓘ http://orcid.org/0000-0002-9178-2487
Yanjun Li ⓘ http://orcid.org/0000-0002-3688-5758

## AUTHOR CONTRIBUTIONS

Tiange Sun, Formal analysis, Visualization, Writing – original draft | Yongtai Li, Supervision, Writing – review and editing | Yuanjing Li, Supervision | Ruixiang Yuan, Supervision | Feng Liu, Supervision | Xiaomei Ma, Supervision | Xinyu Zhang, Supervision | Jie Sun, Supervision.

## DATA AVAILABILITY

The proteomics data in this article can be found in the proteomics database in iProX with the login number of: PXD073036 (https://www.iprox.cn/page/project.html?id=IPX0015162000).

## ADDITIONAL FILES

The following material is available online.

### Supplemental Material

**Fig. S1A (Spectrum02761-25-s0001.tif).** Fig. S1A
**Fig. S1B (Spectrum02761-25-s0002.docx).** Fig. S1B: Heatmap showing the expression fold change of SCRPs in different comparisons.
**Fig. S1C (Spectrum02761-25-s0003.docx).** Fig. S1C Heatmap showing the expression fold change of N-SCPs in different comparisons.
**Table S1 (Spectrum02761-25-s0004.docx).** Table S1: Primer sequences.
**Table S2 (Spectrum02761-25-s0005.docx).** Table S2: The list of PHI proteins identified from 94 SPs.

### Open Peer Review

**PEER REVIEW HISTORY (review-history.pdf).** An accounting of the reviewer comments and feedback.

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
