## [Reviewer comments · Microbiology Spectrum]

Microbiology Spectrum

The deletion of a major facilitator superfamily gene *VdMFS2* results in enhanced pathogenicity of *Verticillium dahliae* to cotton

Yuqiang Sun, Tiange Sun, Yongtai Li, Yuanjing Li, Ruixiang Yuan, Feng Liu, Xinyu Zhang, Jie Sun, Xiaomei Ma, and Yanjun Li

Corresponding Author(s): Yanjun Li, The Key Laboratory of Oasis Eco-Agriculture, Agriculture College, Shihezi University

Review Timeline:

Submission Date:	September 10, 2025
Editorial Decision:	October 22, 2025
Revision Received:	January 20, 2026
Accepted:	February 23, 2026

Editor: Yuheng Yang

Reviewer(s): Disclosure of reviewer identity is with reference to reviewer comments included in decision letter(s). The following individuals involved in review of your submission have agreed to reveal their identity: xiaofeng su (Reviewer #2)

Transaction Report:

DOI: <https://doi.org/10.1128/spectrum.02761-25>

Re: Spectrum02761-25 (**The deletion of a major facilitator superfamily gene *VdMFS2* results in enhanced pathogenicity of *Verticillium dahliae* to cotton**)

Dear Prof. Yuqiang Sun:

Thank you for the privilege of reviewing your work. Below you will find my comments, instructions from the Spectrum editorial office, and the reviewer comments.

Revision Guidelines

Sincerely,
Yuheng Yang
Editor
Microbiology Spectrum

Reviewer #1 (Comments for the Author):

This manuscript is titled "The deletion of a major facilitator superfamily gene *VdMFS2* results in enhanced pathogenicity of *Verticillium dahliae* to cotton" reported reports that *VdMFS2* is a negative regulator of the growth, development and stress resistance of *V. dahliae*. The authors showed that *VdMFS2* mutation accelerated colony growth rates, enhanced carbon source utilization ability, reduced leucine and valine utilization ability, increased stress tolerance, increased host penetration ability and

increased virulence. Nonetheless, there are several noticeable shortcomings in the current manuscript that need to be addressed to strengthen this work.

My major concern is the mechanistic links between increased fungal growth and the observed phenotypic traits, including increased pathogenicity, improved stress tolerance, and increased secreted protein production. MS fails to clarify whether these downstream phenotypes are direct results of VdMFS2 deletion or indirect effects derived from increased fungal biomass.

Minors

lines 11-12, "It was found VdMFS2 silencing significantly reduced cotton resistance to *Verticillium wilt*." In this study, there was no evidence that VdMFS2 silencing reduced cotton resistance but increased the virulence of *V. dahliae*.

More pTRV2:VdMFS2 lines and Δ VdMFS2-C strains should be shown.

In figure 3B, 4C, 4D, 5B, 6B, 7B, 8B, marking different letters to indicate significant differences, which makes these figures very confusing.

Reviewer #2 (Comments for the Author):

This study demonstrates clear scientific significance, with a rigorous experimental design and sufficient data volume, particularly showing considerable depth in the proteomic analysis of the secretome and the investigation of pathogenic mechanisms. However, the manuscript requires major revisions, as there are numerous issues regarding logical flow, data interpretation, figure presentation, and language expression.

1. The text focuses on the role of the MFS family in *Fusarium graminearum* pathogenicity. It is recommended to supplement this by mentioning whether similar functions for this gene family have been reported in *Verticillium dahliae*, to enhance the relevance of the background introduction.

2. Lines 125-126: A 12-hour dark period post-inoculation is stated. However, a 24-hour period is typically used in this research system. The authors should justify the choice of 12 hours and are advised to cite relevant literature supporting this condition.

3. There has an erroneous literature citation in Lines 164-165. The authors should correct this citation and are advised to systematically check all references throughout the manuscript for accuracy.

4. The notation 'cfu/mL', is inconsistent with the format used elsewhere in the text (e.g., 'CFU/mL'). Please check and standardize the notation throughout the manuscript.

5. Materials and Methods sections 2.6 and 2.8 clearly mentioned 'colony diameter was measured at 15 days post-inoculation', but Results section 3.3 described 'the Δ VdMFS2 mutant colonies were significantly larger than the wild-type after 20 days of cultivation. Similarly, Results section 3.4 mentioned 'disease index was assessed at 14 and 21 days post-inoculation (dpi)', but the legend for Figure 4 described '14 and 28 dpi.' The conflicting time points cause confusion.

6. In Materials and Methods section 2.8 (Nitrogen utilization assay), the author described 'the control was Czapek-Dox medium without a carbon source'. This is logically flawed, as the control for a nitrogen source experiment should be 'medium without a nitrogen source.'

7. The legends of Figure 3 is 'VdMFS2 deletion accelerates hyphal growth rate in *V. dahliae*' (focusing on hyphal growth rate). However, Figure 3 also concluded colony morphology (Figure 3A), colony diameters (Figure 3B) and conidial production (Figure 3D). The title does not fully cover the core metrics presented and may mislead readers.

8. Why did the authors culture *V. dahliae* on three different media CM, Czapek, and PDA showed in Figure 3A? What are the differences between these media? In Figure 3C, why did the authors use PDA medium. However, they used Czapek medium in Figure 3D. Why did they use different media in the two experiments?

9. Figure 4: In Figure 4A, what do 'Above' and 'Below' represent? Additionally, the images of stem dissection require scale bars to objectively demonstrate the degree of vascular browning.

10. The authors investigated the sugar transport function of VdMFS2 by using the yeast deletion mutant EBY.VW4000. However, the manuscript fails to describe the specific genetic background of this mutant, such as which specific hexose transporter genes are deleted. The author need to add relevant background information describing the EBY.VW4000 strain in the Materials and Methods section.

11. Results section 3.10 mentioned '18 SCRPs were identified,' but the heatmap in Supplementary Material Fig. S1A lists only 15 SCRPs (e.g., G2WTI6, G2WVX3). Additionally, the author need to perform homology searches for each SCRP and indicate whether they are homologous to previously reported SCRPs known to be associated with *V. dahliae* pathogenicity (e.g., VdSCP41) to enhance the relevance of the findings.

12. 'Host-Induced Gene Silencing (HIGS)' is presented in full capitalized form in the abstract, but written as 'Host-induced gene silencing' in the Results 3.2, with inconsistent usage later. 'non-canonically secreted proteins' is sometimes abbreviated as 'N-CSPs' and sometimes written in full, creating formatting inconsistency. When a term first appears, the author need to indicate the 'full term (abbreviation)' and subsequently use the abbreviation consistently (e.g., HIGS, N-CSPs). Please check the consistency of other key abbreviations such as MFS, CAZymes, SCRPs, and CWDEs.

13. In the Discussion, the authors described relevant studies and the importance of SCRPs by citing several reports. Were all these mentioned SCRPs found among the differentially expressed proteins in this study? Please clarify which of the differentially expressed SCRPs identified in this study have been previously reported.

14. Transcriptome analysis revealed altered expression of a large number of cell wall-degrading enzyme (CWDE) genes in the mutant. The authors should discuss whether there are any reports that deletion of reported MFS family genes has been shown to affect the expression of CWDE genes. Furthermore, to establish a causal relationship, it is recommended to consider supplementing the study with Host-Induced Gene Silencing (HIGS) experiments to further validate the role of these genes in the pathogenic process.

This study demonstrates clear scientific significance, with a rigorous experimental design and sufficient data volume, particularly showing considerable depth in the proteomic analysis of the secretome and the investigation of pathogenic mechanisms. However, the manuscript requires major revisions, as there are numerous issues regarding logical flow, data interpretation, figure presentation, and language expression.

1. The text focuses on the role of the MFS family in *Fusarium graminearum* pathogenicity. It is recommended to supplement this by mentioning whether similar functions for this gene family have been reported in *Verticillium dahliae*, to enhance the relevance of the background introduction.

2. Lines 125–126: A 12-hour dark period post-inoculation is stated. However, a 24-hour period is typically used in this research system. The authors should justify the choice of 12 hours and are advised to cite relevant literature supporting this condition.

3. There has an erroneous literature citation in Lines 164–165. The authors should correct this citation and are advised to systematically check all references throughout the manuscript for accuracy.

4. The notation 'cfu/mL', is inconsistent with the format used elsewhere in the text (e.g., 'CFU/mL'). Please check and standardize the notation throughout the manuscript.

5. Materials and Methods sections 2.6 and 2.8 clearly mentioned 'colony diameter was measured at 15 days post-inoculation', but Results section 3.3 described 'the $\Delta VdMFS2$ mutant colonies were significantly larger than the wild-type after 20 days of cultivation. Similarly, Results section 3.4 mentioned 'disease index was assessed at 14 and 21 days post-inoculation (dpi)', but the legend for Figure 4 described '14 and 28 dpi.' The conflicting time points cause confusion.

6. In Materials and Methods section 2.8 (Nitrogen utilization assay), the author described 'the control was Czapek-Dox medium without a carbon source'. This is logically flawed, as the control for a nitrogen source experiment should be 'medium without a nitrogen source.'

7. The legends of Figure 3 is 'VdMFS2 deletion accelerates hyphal growth rate in *V. dahliae*' (focusing on hyphal growth rate). However, Figure 3 also concluded colony morphology (Figure 3A), colony diameters (Figure 3B) and conidial production (Figure 3D). The title does not fully cover the core metrics presented and may mislead readers.

8. Why did the authors culture *V. dahliae* on three different media CM, Czapek, and PDA showed in Figure 3A? What are the differences between these media? In Figure 3C, why did the authors use PDA medium. However, they used Czapek medium in Figure 3D. Why did they use different media in the two experiments?

9. Figure 4: In Figure 4A, what do 'Above' and 'Below' represent? Additionally, the images of stem

dissection require scale bars to objectively demonstrate the degree of vascular browning.

10. The authors investigated the sugar transport function of VdMFS2 by using the yeast deletion mutant EBY.VW4000. However, the manuscript fails to describe the specific genetic background of this mutant, such as which specific hexose transporter genes are deleted. The author need to add relevant background information describing the EBY.VW4000 strain in the Materials and Methods section.

11. Results section 3.10 mentioned '18 SCRPs were identified,' but the heatmap in Supplementary Material Fig. S1A lists only 15 SCRPs (e.g., G2WTI6, G2WVX3). Additionally, the author need to perform homology searches for each SCRPs and indicate whether they are homologous to previously reported SCRPs known to be associated with *V. dahliae* pathogenicity (e.g., VdSCP41) to enhance the relevance of the findings.

12. 'Host-Induced Gene Silencing (HIGS)' is presented in full capitalized form in the abstract, but written as 'Host-induced gene silencing' in the Results 3.2, with inconsistent usage later. 'non-canonically secreted proteins' is sometimes abbreviated as 'N-CSPs' and sometimes written in full, creating formatting inconsistency. When a term first appears, the author need to indicate the 'full term (abbreviation)' and subsequently use the abbreviation consistently (e.g., HIGS, N-CSPs). Please check the consistency of other key abbreviations such as MFS, CAZymes, SCRPs, and CWDEs.

13. In the Discussion, the authors described relevant studies and the importance of SCRPs by citing several reports. Were all these mentioned SCRPs found among the differentially expressed proteins in this study? Please clarify which of the differentially expressed SCRPs identified in this study have been previously reported.

14. Transcriptome analysis revealed altered expression of a large number of cell wall-degrading enzyme (CWDE) genes in the mutant. The authors should discuss whether there are any reports that deletion of reported MFS family genes has been shown to affect the expression of CWDE genes. Furthermore, to establish a causal relationship, it is recommended to consider supplementing the study with Host-Induced Gene Silencing (HIGS) experiments to further validate the role of these genes in the pathogenic process.

Dear Editor,

Thank you very much for your prompt response and for forwarding the reviewers' comments on our manuscript entitled "The deletion of a major facilitator superfamily gene *VdMFS2* results in enhanced pathogenicity of *Verticillium dahliae* to cotton" (Spectrum02761-25). We appreciate the constructive suggestions, which have been very helpful for revising the manuscript and for guiding our future research.

We have carefully considered every comment and have made the corresponding revisions (all changes are highlighted in red in the revised file). We hope that the revised version will now meet the requirements of the journal and look forward to your further evaluation.

Our responses to Reviewer #1's comments are as follows:

1: Reviewer #1's major concern is the mechanistic links between increased fungal growth and the observed phenotypic traits, including increased pathogenicity, improved stress tolerance, and increased secreted protein production. MS fails to clarify whether these downstream phenotypes are direct results of *VdMFS2* deletion or indirect effects derived from increased fungal biomass.

Response: We thank the reviewer for this insightful comment. Although the $\Delta VdMFS2$ mutant grows faster, its enhanced virulence, stress tolerance and increased secreted-protein abundance are not simply a consequence of greater biomass. When equal amounts of total secreted protein from $\Delta VdMFS2$ and WT-Vd592 (normalized to fungal dry weight) were infiltrated into cotton leaves, the $\Delta VdMFS2$ fraction elicited a markedly stronger immune response. Proteomic and enzyme-activity analyses further revealed compensatory up-regulation of key secreted proteins such as CAZymes and SCRPs that act as the principal drivers of virulence. These data demonstrate that *VdMFS2* directly orchestrates metabolism, protein secretion and pathogenicity, rather than merely affecting fungal biomass.

2: Reviewer #1's comments lines 11-12, "It was found *VdMFS2* silencing significantly reduced cotton resistance to *Verticillium* wilt." In this study, there was no evidence that *VdMFS2* silencing reduced cotton resistance but increased the virulence of *V. dahliae*.

Response: We sincerely thank the reviewer for this valuable and insightful comment. We fully agree that the original statement "silencing of *VdMFS2* significantly decreased cotton resistance to *Verticillium* wilt" was inaccurate and potentially misleading. Upon re-examination, we realize that our data only demonstrate that loss of *VdMFS2* increases the pathogen's virulence; they do not provide direct evidence that the host's (cotton's) defense mechanisms are suppressed. The aggravated disease symptoms observed in the experiment resulted from enhanced infectivity of WT-Vd592 following *VdMFS2* silencing, not necessarily indicating a diminished disease resistance in cotton itself. Therefore, we have

revised all relevant sentences throughout the manuscript to reflect this mechanistic distinction. For example, the original sentence now reads: “*VdMFS2* silencing enhances the pathogenicity of *V. dahliae* to cotton.” See lines 11–12 and line 332. These changes align the text with our experimental design and data interpretation. We are grateful to the reviewer for the meticulous correction, which has markedly improved the accuracy and scientific rigor of our work.

3: Reviewer’s comment more pTRV2:*VdMFS2* lines and Δ *VdMFS2-C* strains should be shown.

Response: Thank the reviewer for the valuable comments. We have now included one additional independent replicate for both the pTRV2:*VdMFS2* line and the Δ *VdMFS2-C* complementation strain to address this concern. In Fig.2, Fig.3, Fig.4, Fig.5, Fig.6, Fig.7, and Fig.8. Enhance the credibility of the experimental results.

4: Reviewer’s comment in figure 3B, 4C, 4D, 5B, 6B, 7B, 8B, marking different letters to indicate significant differences, which makes these figures very confusing.

Response: We sincerely apologize for the valuable suggestion regarding the clarity of statistical labeling. We have simplified the letter codes in Figures 3B, 4C, 4D, 5B, 6B, 7B and 8B by replacing overlapping letters with asterisks “*” for $p < 0.05$, “**” for $p < 0.01$ and “***” for $p < 0.001$.

Our responses to Reviewer #2’s comments are as follows:

1: Reviewer’s comment the text focuses on the role of the MFS family in *F. graminearum* pathogenicity. It is recommended to supplement this by mentioning whether similar functions for this gene family have been reported in *V. dahliae*, to enhance the relevance of the background introduction.

Response: We thank the reviewer for this helpful suggestion. We have now identified a report demonstrating that deletion of the MFS-family gene *VdPAT1* in *V. dahliae* attenuates its pathogenicity to cotton (Stephen et al., 2025). The corresponding citation has been added to the revised manuscript in lines 67–68.

2 : Reviewer’s comment lines 125-126: A 12-hour dark period post-inoculation is stated. However, a 24-hour period is typically used in this research system. The authors should justify the choice of 12 hours and are advised to cite relevant literature supporting this condition.

Response: We are deeply sorry for spotting this error. The “12-hour dark period” cited in lines 125–126 was indeed a typo. In our HIGS protocol the correct dark incubation time is 24

h, consistent with the established procedure for cotton (Ratcliff et al., 2001). We have corrected this mistake in the revised manuscript in line 123 and added the reference to the seminal paper that originally defined 24 h darkness as the optimal condition for synchronizing viral infection.

3: Reviewer's comment there has an erroneous literature citation in lines 164-165. The authors should correct this citation and are advised to systematically check all references throughout the manuscript for accuracy.

Response: We sincerely apologize for the incorrect citation on lines 164–165. We have now conducted a systematic check of every reference in the manuscript and corrected the formatting of all literature citations to ensure full compliance with the journal's requirements.

4: Reviewer's comment the notation "cfu/mL", is inconsistent with the format used elsewhere in the text (e.g., "CFU/mL"). Please check and standardize the notation throughout the manuscript.

Response: We thank the reviewer for pointing this out. We have carefully checked the entire manuscript and changed all instances of "cfu/mL" to "CFU/mL" for consistency.

5: Reviewer's comment Materials and Methods sections 2.6 and 2.8 clearly mentioned "colony diameter was measured at 15 days post-inoculation", but Results section 3.3 described" the $\Delta VdMFS2$ mutant colonies were significantly larger than the wild-type after 20 days of cultivation. Similarity, Results section 3.4 mentioned "disease index was assessed at 14 and 21 days post-inoculation (dpi)", but the legend for Figure 4 described "14 and 28 dpi." The conflicting time points cause confusion.

Response: We apologize for the inaccuracies. In Results section 3.3 we incorrectly stated that " $\Delta VdMFS2$ colonies were significantly larger than the wild-type after 20 days of culture"; likewise, section 3.4 referred to disease-index evaluation at "14 and 21 days post-inoculation (dpi)." These were typographical errors. In our experimental protocol the correct incubation time for colony growth is 15 days, and disease indices were recorded at 14 and 28 dpi. We have corrected these mistakes in the revised manuscript in lines 324 and 390.

6: Reviewer's comment in Materials and Methods section 2.8 (Nitrogen utilization assay), the author described "the control was Czapek-Dox medium without a carbon source". This is logically flawed, as the control for a nitrogen source experiment should be 'medium without a nitrogen source.'

Response: We thank the reviewer for identifying this logical error. We have revised "Czapek-Dox medium without a carbon source" to "Czapek-Dox medium without a nitrogen source" in line 211-212.

7 : Reviewer’s comment the legends of Figure 3 is “VdMFS2 deletion accelerates hyphal growth rate in *V. dahlia*” (focusing on hyphal growth rate). However, Figure 3 also concluded colony morphology (Figure 3A), colony diameters (Figure 3B) and conidial production (Figure 3D). The title does not fully cover the core metrics presented and may mislead readers.

Response: We thank the reviewer for this comment. The original title “*VdMFS2* deletion accelerates hyphal growth rate in *V. dahliae*.” was indeed too narrow. We have revised it to “*VdMFS2* deletion accelerated the growth of *V. dahliae*” to better reflect the overall growth-promoting effect observed in line 363.

8: Reviewer’s comment why did the authors culture *V. dahliae* on three different media CM, Czapek, and PDA showed in Figure 3A? What are the differences between these media? In Figure 3C, why did the authors use PDA medium. However, they used Czapek medium in Figure 3D. Why did they use different media in the two experiments?

Response: The three media serve distinct purposes and differ in composition as follows (1) CM (Complete Medium): Mimics the nutrient-rich conditions of a parasitic environment; used to assess basal growth differences between wild-type and mutant under ample nutrients. (2) Czapek-Dox (Defined Medium): Contains only 0.02 % KNO₃ as the nitrogen source. Its low-nitrogen formulation amplifies metabolic defects, making it ideal for detecting amino-acid transport deficiencies and their impact on nitrogen utilization. (3) PDA (Potato-Dextrose Agar): Rich in plant polysaccharides and trace elements; employed to evaluate fungal adaptability under host-mimicking conditions. Figure 3C (PDA): Focuses on colony expansion rate. The high carbohydrate content (20 % potato extract) promotes rapid mycelial spread, facilitating quantitative measurement of growth kinetics (Klosterman et al., 2009). Figure 3D (Czapek): Conidiation is sensitive to nitrogen availability. The stringent nitrogen limitation (0.02 % KNO₃) in Czapek medium specifically induces sporulation while avoiding the suppressive effects of high nitrogen (Czapek, 1902), and this point has been further clarified in the Materials and Methods, section 2.6 (lines 178–186).

9: Reviewer’s comment figure 4: In Figure 4A, what do “Above” and “Below” represent? Additionally, the images of stem dissection require scale bars to objectively demonstrate the degree of vascular browning.

Response: In the original figure, “Above” and “Below” were intended to indicate the front and back sides of the medium, whereas “Before” and “After” refer to the states before and after the cellophane was peeled off. To convey the intended meaning of “before and after peeling off the cellophane,” we have replaced “Above” and “Below” with “Before” and “After.” We have also added scale bars to the vascular-bundle images to improve accuracy in

figure 4.

10: Reviewer’s comment the authors investigated the sugar transport function of *VdMFS2* by using the yeast deletion mutant EBY.VW4000. However, the manuscript fails to describe the specific genetic background of this mutant, such as which specific hexose transporter genes are deleted. The author need to add relevant background information describing the EBY.VW4000 strain in the Materials and Methods section.

Response: We thank the reviewer for pointing out that the genetic background of EBY.VW4000 was not described in detail in the Materials and Methods. Following your suggestion, we have added the following information to the revised manuscript: “EBY.VW4000 was constructed by Wieczorke et al. (1999). Its genome lacks all 18 hexose-transporter genes (HXT1–HXT17 and GAL2) and the two glucose sensors SNF3 and RGT2; consequently, it cannot grow on media containing hexoses (glucose, fructose, mannose, etc.) as the sole carbon source and is widely used as a reporter strain for functional studies of hexose transporters.” In line 217-223.

11: Reviewer’s comment results section 3.10 mentioned '18 SCRPs were identified,' but the heatmap in Supplementary Material Fig. S1A lists only 15 SCRPs (e.g., G2WTI6, G2WVX3). Additionally, the author need to perform homology searches for each SCRPs and indicate whether they are homologous to previously reported SCRPs known to be associated with *V. dahliae* pathogenicity (e.g., VdSCP41) to enhance the relevance of the findings.

Response: We apologize for the oversight; we have now corrected the number of SCRPs listed in Figure S1 from 15 to 18. We performed a homology comparison of these 18 SCRPs and found that two of them have been previously reported to be associated with the pathogenicity of *V. dahliae*: G2WWQ6, a member of the classic fungal effector family *VdCPI*, which can trigger plant immunity (Zhang et al., 2020); G2WYQ1, a secreted glycosidase *VdGLU1*, involved in cell wall degradation and enhancement of virulence by inducing ER-stress (Wang et al., 2022). The remaining SCRPs have not been reported in the literature. We have now highlighted these two previously characterized SCRPs in the Results section in line 575-578.

12: Reviewer’s comment “Host-Induced Gene Silencing (HIGS)” is presented in full capitalized form in the abstract, but written as “Host-induced gene silencing” in the Results 3.2, with inconsistent usage later. “non-canonically secreted proteins” is sometimes abbreviated as “N-CSPs” and sometimes written in full, creating formatting inconsistency. When a term first appears, the author need to indicate the “full term (abbreviation)” and subsequently use the abbreviation consistently (e.g., HIGS, N-CSPs). Please check the consistency of other key abbreviations such as MFS, CAZymes, SCRPs,

and CWDEs.

Response: We sincerely apologize for this mistake. We have changed “Host-induced gene silencing” to “HIGS” in section 3.2 and have checked and corrected all key abbreviations throughout the manuscript.

13: Reviewer’s comment in the Discussion, the authors described relevant studies and the importance of SCRPs by citing several reports. Were all these mentioned SCRPs found among the differentially expressed proteins in this study? Please clarify which of the differentially expressed SCRPs identified in this study have been previously reported.

Response: We sincerely thank the reviewer for raising this important issue. A homology search across the 18 SCRPs revealed that two have previously been implicated in *Verticillium dahliae* pathogenicity: G2WWQ6, a member of the canonical fungal effector family *VdCPI* that triggers plant immunity (Zhang et al., 2020), and G2WYQ1, a secreted glycosidase (*VdGLU1*) that enhances virulence via cell-wall degradation and ER-stress induction (Wang et al., 2022). The remaining SCRPs have not been functionally characterized to date; these two known proteins are now highlighted in the Results in line 688-691.

14: Reviewer’s comment transcriptome analysis revealed altered expression of a large number of cell wall-degrading enzyme (CWDE) genes in the mutant. The authors should discuss whether there are any reports that deletion of reported MFS family genes has been shown to affect the expression of CWDE genes. Furthermore, to establish a causal relationship, it is recommended to consider supplementing the study with Host-Induced Gene Silencing (HIGS) experiments to further validate the role of these genes in the pathogenic process.

Response: We sincerely thank the reviewer for this valuable suggestion. As reported for *FgMFS1*, that transporter regulates deoxynivalenol (DON) export, and its deletion reduces expression of pectinase (*pel*) and cellulase (*cel*) genes by more than 50%, impairing host cell-wall degradation. Following this rationale, we selected the hemicellulase G2WZT4 (α -glucuronidase) identified among the carbohydrate hydrolases, performed HIGS silencing, and found that it significantly attenuates the virulence of *V. dahliae* in figure S1 (A).

Once again, we sincerely thank both reviewers and the editor for their meticulous review and invaluable suggestions. We believe that the revised manuscript is now more rigorous and complete, and that it meets the requirements for publication in your journal.

With highest regards,

The Author Team

Re: Spectrum02761-25R1 (**The deletion of a major facilitator superfamily gene *VdMFS2* results in enhanced pathogenicity of *Verticillium dahliae* to cotton**)

Dear Prof. Yanjun Li:

Your manuscript has been accepted, and I am forwarding it to the ASM production staff for publication. Your paper will first be checked to make sure all elements meet the technical requirements. ASM staff will contact you if anything needs to be revised before copyediting and production can begin. Otherwise, you will be notified when your proofs are ready to be viewed.

Sincerely,
Yuheng Yang
Editor
Microbiology Spectrum